# Polypharmacology guided drug repositioning approach for SARS-CoV2

Esther Jamir[1,2]*, Himakshi Sarma[1], Lipsa Priyadarsinee[1,2], Kikrusenuo Kiewhuo[1,2], Selvaraman Nagamani[1,2], G. Narahari Sastry[1,2]*

1 Advanced Computation and Data Sciences Division, CSIR–North East Institute of Science and Technology, Jorhat, Assam, India, 2 Academy of Scientific and Innovative Research (AcSIR), Ghaziabad, India

* gnsastry@gmail.com, gnsastry@neist.res.in (GNS); essjmr@gmail.com (EJ)

**Data Availability Statement:** All relevant data are within the paper and its Supporting Information files.

## Abstract

Drug repurposing has emerged as an important strategy and it has a great potential in identifying therapeutic applications for COVID-19. An extensive virtual screening of 4193 FDA approved drugs has been carried out against 24 proteins of SARS-CoV2 (NSP1-10 and NSP12-16, envelope, membrane, nucleoprotein, spike, ORF3a, ORF6, ORF7a, ORF8, and ORF9b). The drugs were classified into top 10 and bottom 10 drugs based on the docking scores followed by the distribution of their therapeutic indications. As a result, the top 10 drugs were found to have therapeutic indications for cancer, pain, neurological disorders, and viral and bacterial diseases. As drug resistance is one of the major challenges in antiviral drug discovery, polypharmacology and network pharmacology approaches were employed in the study to identify drugs interacting with multiple targets and drugs such as dihydroergotamine, ergotamine, bisdequalinium chloride, midostaurin, temoporfin, tirilazad, and venetoclax were identified among the multi-targeting drugs. Further, a pathway analysis of the genes related to the multi-targeting drugs was carried which provides insight into the mechanism of drugs and identifying targetable genes and biological pathways involved in SARS-CoV2.

## Introduction

The SARS-CoV2 (severe acute respiratory syndrome coronavirus 2) has caused a pandemic [1–3] and it has affected the lives of millions of people severely collapsing the healthcare system and economy [4]. The development of therapeutics and antiviral drugs is critical for meeting the challenges currently faced by viral infections [5]. Though therapies are available for the treatment of SARS-CoV2 infection, they deal with only the symptoms of the disease but not its underlying causes and have not been able to eradicate the disease completely [6–9]. The emergence of new mutant strains may be the primary cause of the increased transmissibility of SARS-CoV2 virus [10, 11]. So far, approximately 30 SARS-CoV2 proteins have been identified [12], and the occurrence of mutations have increased the rate of infection, replication, transcription and transmission [13–15]. SARS-CoV2 is reported to be 96.2% similar to bat CoV RaTG13 sequence and it shares 79.5% identity with SARS-CoV genome [16]. A 5' cap and a 3' poly-A tail is present in the positive-stranded RNA genome structure [17] where a replicase

**Funding:** Funding to carry out the research work was obtained by DBT Centre of Excellence in Advanced Computation and Data Sciences (No. BT/PR40188/BTIS/137/27/2021).

**Competing interests:** The authors declare no competing interests.

complex of ORFs i.e., ORF1a and ORF1b are present at the 5' cap. The ORF1a and ORF1b produces polypeptide pp1a and pp1ab respectively, the pp1a comprises non-structural protein (NSP) 1 to NSP11, while pp1ab comprises of NSP12 to NSP16. These NSPs are involved in viral replication, transcription and immune invasion [18]. Different structural proteins such as S (Spike), N (Nucleoprotein), M (Membrane), and E (Envelope) proteins are encoded at the 3' end [19]. There are nine accessory proteins (ORF3a, ORF3b, ORF6, ORF7a, ORF7b, ORF8, ORF9b, ORF9c, ORF10). These proteins are significant for virulence and host interaction [20–22]. Due to the high sequence similarity among the coronavirus, these proteins could be studied for the development of an antiviral drug. **S1a, S1b Fig in S1 File** displays the selected proteins of SARS-CoV2 that are involved in viral entry, RNA synthesis and viral replication.

The drug and vaccine development process for the treatment and prevention of COVID-19 has intrigued the scientific community's interest around the world [23, 24]. Several vaccines have been developed around the world, but due to the rise of new viral strains, it is still a challenge to eradicate the virus [25]. Thus, the drug repurposing [26, 27] approach may have the possibility to identify the potential drug molecules against COVID-19 within a short period. Numerous modelling approaches have been applied for repurposing pre-existing drugs [28–30] to halt the rise of the COVID-19 pandemic. Although there are a few repurposed drugs for COVID-19 available in the market, some of the drugs are reported to have limitations. For instance, patients under remdesivir treatment were reported to be experiencing adverse events during follow-up. Similarly, few of the repurposed drugs for COVID-19 such as hydroxychloroquine and chloroquine were reported to cause serious heart problems and other safety issues [31, 32]. Several approaches such as polypharmacology with drug repurposing are reported to be effective for exploring potential leads and druggable targets for infectious diseases [33]. The polypharmacology approach involves the identification of small molecules that have activity against multiple targets. This method is widely used for identifying drugs with polypharmacological properties for targeting multiple SARS-CoV2 proteins [34]. In our previous study, a combined polypharmacology and drug repurposing approach was applied to screen FDA (Food and Drug Administration) approved drugs against seven SARS-CoV2 targets. Four drugs (i.e., venetoclax, tirilazad, acetyldigitoxin, and ledipasvir) were chosen based on the docking scores, binding pose and the interaction pattern with the SARS-CoV2 targets [35]. In arriving at the drug repurposing a consensus scoring approach has been adopted [36]. Similarly, earlier in the group, we employed an integrated approach involving structure and analog-based approaches to arrive at lead molecules [37–40]. All 30 targets were carefully analyzed and 24 targets were selected based on their functions, binding pocket size and the presence of active site residues. The 24 targets considered in this study are involved in viral entry, RNA synthesis, polyprotein cleavage and replication.

## Materials and methods

### Selection of target proteins and their structural details

In this study, 24 SARS-CoV2 protein targets were considered based on their role in viral entry, replication, transcription, presence of a binding pocket and the availability of protein structure with a good resolution in RCSB Protein Data Bank (PDB) (http://www.rcsb.org) [41]. **Table 1** provides the PDB ID, resolutions, length, and activity of the chosen targets: NSP1, NSP2, NSP3, NSP5, NSP7, NSP8, NSP9, NSP10, NSP12, NSP13, NSP14, NSP15, NSP16, S, E, N, ORF3a, ORF7a, ORF8, and ORF9b, NSP4, NSP6, M, and ORF6. Among these, four proteins namely NSP4, NSP6, M and ORF6 were obtained from the I-TASSER server (https://zhanggroup.org/I-TASSER/) [42] due to the lack of available resolved crystal structures during this study. The 3D structures of the proteins are given in **S1a, S1b Fig in S1 File**.

**Table 1. List of 24 SARS-CoV2 targets along with their PDB ID, resolution, sequence length and function.**

| Sl. no | PDB ID | Name of the target | Resolution | Length | Function |
|---|---|---|---|---|---|
| 1 | 7K7P | Non-Structural Protein 1 | 1.77 | 118 | Host cell modulation |
| 2 | 7EXM | Non-Structural Protein 2 | 1.74 | 278 | Cell degradation |
| 3 | 7LOS | Non-Structural Protein 3 (PLpro) | 2.9 | 316 | Viral replication |
| 4 | Modelled | Non-Structural Protein 4 | - | 499 | Cleaves the C-terminus of replicase polyprotein at 11 sites |
| 5 | 6LU7 | Non-Structural Protein 5 (3Clpro) | 2.16 | 30 6 | Recognizes substrates containing the core sequence |
| 6 | Modelled | Non-Structural Protein 6 | - | 290 | Initial induction of autophagosomes from host reticulum endoplasmic |
| 7 | 6M5I | Non-Structural Protein 7 | 2.5 | 198 | Viral replication |
| 8 | 7CYQ | Non-Structural Protein 8 | 2.83 | 942 | Viral replication |
| 9 | 6W4B | Non-Structural Protein 9 | 2.95 | 117 | Viral replication |
| 10 | 6W4H | Non-Structural Protein 10 | 1.80 | 142 | Viral transcription |
| 11 | 6M71 | Non-Structural Protein 12 (RNA-dependent RNA polymerase) | 2.9 | 942 | Replication and transcription of the viral RNA genome |
| 12 | 7NN0 | Non-Structural Protein 13 (Helicase) | 3.04 | 603 | Multi-functional protein having a zinc- binding domain in N-terminus |
| 13 | 7MC5 | Non-Structural Protein 14 (Proofreading exoribonuclease) | 1.64 | 287 | Proofreading exoribonuclease for RNA replication |
| 14 | 6VWW | Non-Structural Protein 15(Uridylate-specific endoribonuclease) | 2.2 | 370 | Cleaves 2'-3'-cyclic phosphates 5' to the cleaved bond |
| 15 | 6W4H | Non-Structural Protein 16 (2'-O-methyltransferase) | 1.8 | 301 | Viral mRNAs cap methylation |
| 16 | 6M0J | Spike glycoprotein (S protein) | 2.45 | 603 | Viral fusion peptide |
| 17 | 5X29 | Envelope protein (E) | - | 81 | Virus morphogenesis and assembly |
| 18 | Modelled | Membrane protein (M) | - | 194 | Virus morphogenesis and assembly |
| 19 | 6WJI | Nucleoprotein (N) | 2.05 | 121 | RNA transcription and viral replication |
| 20 | 6XDC | Open Reading Frame 3a protein | 2.9 | 284 | Up-regulates expression of fibrinogen subunits FGA, FGB and FGG |
| 21 | Modelled | Open Reading Frame 6 protein | - | 61 | Blocks multiple antiviral activities |
| 22 | 6W37 | Open Reading Frame 7a protein | 2.9 | 67 | Virus replication in cell culture |
| 23 | 7JTL | Open Reading Frame 8 protein | 2.04 | 107 | Host-virus interaction |
| 24 | 6Z4U | Open Reading Frame 9b protein | 1.95 | 97 | Inhibits the host innate immune response |

## Preparation of protein structures

The selected 24 SARS-CoV2 proteins were prepared in Autodock tools (ADT version 1.5.7rc1) [43], where the solvent molecules and ions were removed, and the partial charges and polar hydrogens were added to the structure. The protein structures were then saved in the PDBQT format. In addition, a grid box was generated based on the active site and neighbouring residues as shown in **S1 Table in S1 File**.

## Preparation of compound structures

The drug molecules were retrieved from DrugBank [44] and DrugCentral [45] databases. The redundant molecules and molecules containing ions were removed and 4193 unique approved molecules were considered in this study. Further, the raccoon.py script of MGL tools was used for optimizing the geometry of the compounds, and the compounds were converted to 3D structures and PDBQT format.

## Virtual screening

The virtual screening of the 4193 known drugs against the 24 SARS-CoV2 targets was performed using AutodockVina1.1.2. Based on the docking scores the best conformer was

identified out of the five conformers generated and the selected complexes were then taken for further analysis. Further to validate the virtual screening, three other programs namely Molecular Transfer Drug Target Interaction prediction (MT-DTI), SwissDock, and iGEMDOCK was used to calculate the binding energies of the top 10 molecules for each target. The MT-DTI is a drug-target interaction predicting module using a deep learning library DeepPurpose [46]. SwissDock is a web-based program for protein-ligand docking and virtual screening and its scoring function is based on empirical force filed and statistical functions [47]. The iGEMDOCK is a software tool that predicts the binding mechanism and affinities of protein-ligand complexes and rates the screened compounds using pharmacological interactions and an energy-based scoring function [48].

## Analysis of most active and least active molecules

The docking score of all the drug molecules for each target was arranged in ascending order and the frequency of the docking score was analysed. The maximum number of drugs that fall in a range of docking scores were considered as the "threshold" value and the list of drugs was classified into two different portions. The docking scores higher than the threshold were considered active molecules and lesser than the threshold was considered inactive molecules. Further, the top 10 ranked candidates from the active molecule and 10 low-ranking candidates from the inactive molecules were analysed in detail.

## Distribution of therapeutic indications toward the SARS-CoV2 targets

The top 10 and bottom 10 compounds were selected to analyze the most active and least active therapeutic indications for the SARS-CoV2 targets. The therapeutic indication of each compound was retrieved from the drug databases namely DrugBank [44] and DrugCentral databases [45] and then the distribution of drugs with a category of therapeutic indications towards the SARS-CoV2 targets was carried out for the top 10 and bottom 10 compounds. In addition, the reported FDA approved drugs and drugs under clinical trials for COVID-19 disease were identified among the top-scored drugs in the study. **Fig 1** depicts the schematic workflow employed in the current study.

## Results

### Virtual screening of drug molecules against 24 targets

Virtual screening of the known drug molecules obtained from DrugBank [44] and DrugCentral database [45] was carried out against the 24 SARS-CoV2 viral targets. The molecules were distributed based on the docking score and a bar graph was plotted for each target to identify the number of drugs falling under different docking scores (**S2a, S2b Fig in S1 File**). It can be observed that the average maximum docking score for each protein is around -5.0 kcal/mol to -7.0 kcal/mol and the least docking scores were found to be around -1.0 kcal/mol to -4.0 kcal/mol.

Further drug molecules were categorised into active and inactive dataset where the docking score cut-off was determined for each target (i.e., the score range that had maximum no. of drugs) and the drug molecules which scored higher than the cut-off were considered as most active molecules whereas the molecules whose docking score was less than the cut-off were considered as least active molecules. The threshold value was selected based on the highest docking score for each protein distributed across the drug. **Table 2** shows the grouping of the total compounds into active and least active groups based on the cut-off scores. The top ten molecules from the most active groups and the bottom ten compounds from the least active

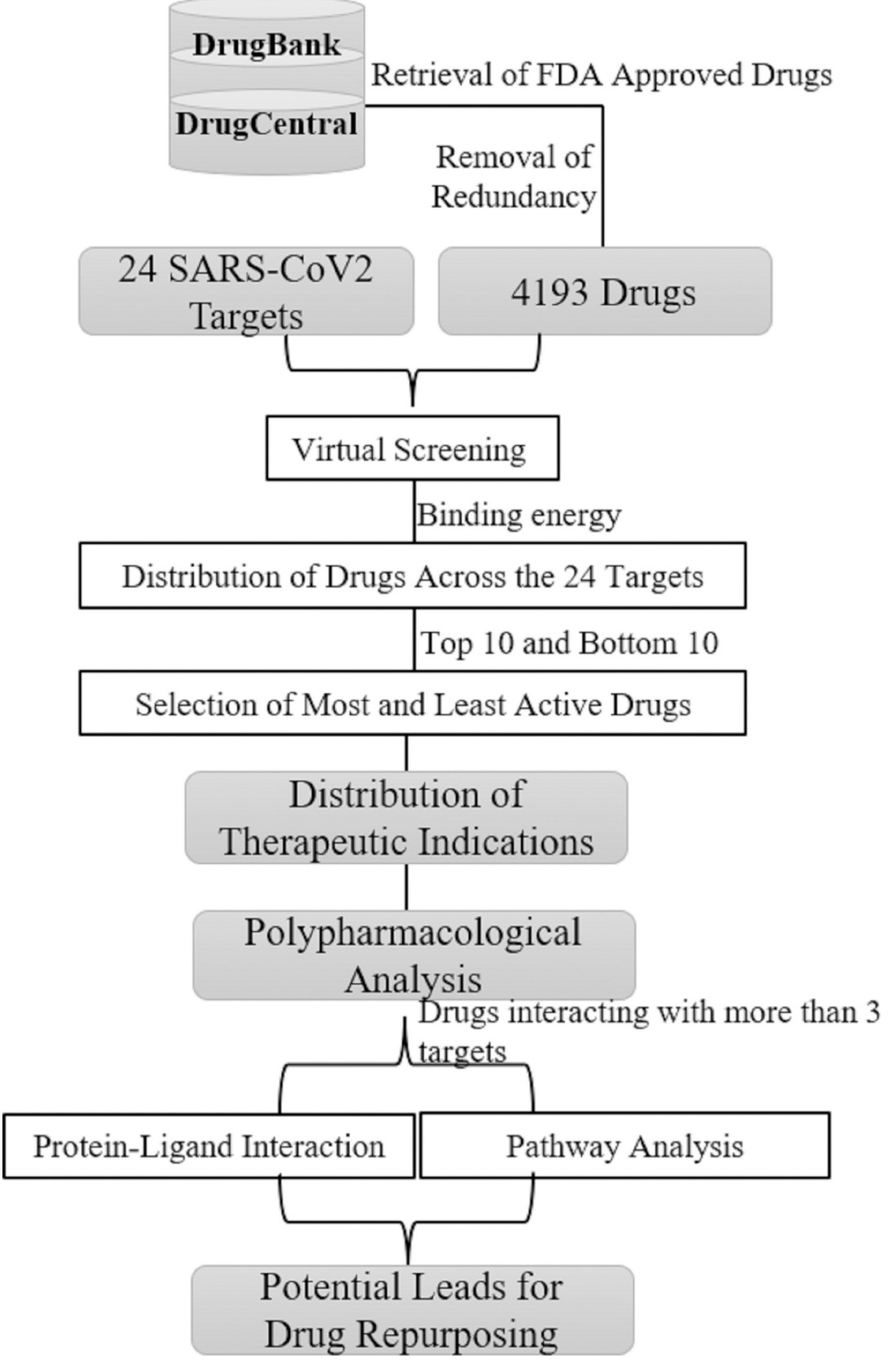

**Fig 1. Schematic workflow of screening known drugs for therapeutic indications against SARS-CoV2 selected targets.**

**Table 2. Distribution of active and inactive drugs across all the 24 SARS-CoV2 proteins.** The total number of drugs that has docking score > threshold value has been considered as active (%) control, whereas the drugs that have docking score < threshold value has been considered as inactive (%).

| Sl.No. | Protein | Inactive (%) | Active (%) | Threshold value (kcal/mol) |
|---|---|---|---|---|
| 1 | Envelope | 42.00 | 58.00 | -4.0 |
| 2 | Membrane | 45.34 | 54.66 | -7.0 |
| 3 | Nucleoprotein | 42.07 | 57.93 | -7.0 |
| 4 | Spike | 38.54 | 61.46 | -5.5 |
| 5 | NSP1 | 60.98 | 39.02 | -5.0 |
| 6 | NSP2 | 44.77 | 55.23 | -7.0 |
| 7 | NSP3 | 61.65 | 38.35 | -7.0 |
| 8 | NSP4 | 63.44 | 36.56 | -6.0 |
| 9 | NSP5 | 43.72 | 56.28 | -6.0 |
| 10 | NSP6 | 35.20 | 64.80 | -5.5 |
| 11 | NSP7 | 44.19 | 55.81 | -3.0 |
| 12 | NSP8 | 48.72 | 51.28 | -5.0 |
| 13 | NSP9 | 59.48 | 40.52 | -6.0 |
| 14 | NSP10 | 28.71 | 71.29 | -6.0 |
| 15 | NSP12 | 41.43 | 58.57 | -6.0 |
| 16 | NSP13 | 25.09 | 74.91 | -6.0 |
| 17 | NSP14 | 43.07 | 56.93 | -6.0 |
| 18 | NSP15 | 49.49 | 50.51 | -6.0 |
| 19 | NSP16 | 50.35 | 49.65 | -6.0 |
| 20 | ORF3a | 40.45 | 59.55 | -5.5 |
| 21 | ORF6 | 34.61 | 65.39 | -5.0 |
| 22 | ORF7a | 74.74 | 25.26 | -4.0 |
| 23 | ORF8 | 49.84 | 50.16 | -4.5 |
| 24 | ORF9b | 48.20 | 51.80 | -5.0 |

groups were selected for further analysis. From the distribution of active and least active drugs for 24 SARS-CoV2 targets, it can be observed that among the total number of drugs, the average percentage of drugs falling under the most active drugs was found to be 53.5% while the average percentage of drugs falling under the least active drugs was found to be 46.5%. The drugs falling under the most and least active drugs were further taken for analysis of therapeutic indications. In addition, to confirm the binding affinity of the drug-target interaction, MT-DTI prediction, SwissDock, and iGEMDOCK was used to validate the top 10 molecules displaying the best docking score against each target. It has been observed that the binding scores obtained through the three programs were comparable with that of the docking energy obtained from Autodock Vina. Several studies have also reported the prediction of antivirals such as remdesivir and lopinavir for SARS-CoV2 using MT-DTI and also the use of SwissDock and iGEMDOCK for screening potential leads against SARS-CoV2 targets [49–51]. Thus, the confirmation of docking scores of top 10 FDA approved drugs across the 24 SARS-CoV2 targets with the binding affinity obtained from MT-DTI model, SwissDock and iGEMDOCK validates the screening and selection of top 10 drugs. The docking and binding scores calculated from Autodock vina, MT-DTI, SwissDock, and iGEMDOCK are listed in **S2 Table in S1 File**. In order to validate the docking protocol, the free energy calculation was carried out using online server FastDRH (http://cadd.zju.edu.cn/fastdrh/submit) for multi targeting drugs. This server integrates Autodock Vina, Autodok GPU docking engines, structure truncated MM/PB (GB)SA, per-residue energy decomposition analysis for multiple residues to predict the

binding free energies. The analysis revealed that the binding free energies are well co-related with docking score (**S3 Table in S1 File**).

## Therapeutic indications of the top 10 and bottom 10 drugs

The therapeutic indication of all the drug molecules was identified through DrugBank [44] and DrugCentral [45] databases which were then analysed to identify drugs with a category of therapeutic indications that may be repurposed against 24 SARS-CoV2 targets. The top 10 drugs that showed good binding affinity towards the 24 SARS-CoV2 targets are listed in **Tables 3–5** along with their existing therapeutic indication for structural, non-structural and accessory SARS-CoV2 proteins respectively. From the table, it can be observed that among a total number of 240 drugs in the top 10, around 162 drugs were unique and 39 drugs occurred multiple times across the 24 SARS-CoV2 targets.

The overall binding score distribution of the top 10 drugs across the 24 SARS-CoV2 targets is given in **Fig 2**. From the graph in **Fig 2A**, among the non-structural proteins, it can be observed that sonidegib, which is an anticancer drug, has the highest docking score of -11.8 kcal/mol with NSP16 protein. And drugs such as desloratadine, mianserin, metapramine and mecloqualone which are used as anti-depressive agents have the lowest docking score of around -5.5 kcal/mol with NSP7 protein. In the case of structural proteins shown in **Fig 2B**, nucleoprotein interacting with paritaprevir drug which is used for the treatment of hepatitis was found to have the highest docking score of -12.6 kcal/mol. While drugs interacting with envelope proteins which showed therapeutic indications for liver disorder, pain, diabetes, nausea, cancer, and pancreas disease appeared to have lower docking score of around -6.0 kcal/mol compared to the other docking score. Then, among the accessory proteins as shown in **Fig 2C**, ORF6 is observed to have the highest docking score of -10.1 kcal/mol with bisdequalinium chloride which is used for bacterial infections. While, the protein ORF7a had a lower docking score of around -5.5 kcal/mol with drugs such as hydromorphone, naproxen, phenformin, primidone which are commonly used for treating pain, haemorrhage and epilepsy.

The overall distribution of therapeutic indications in terms of disease area for the top 10 and bottom 10 drugs across the 24 SARS-CoV2 targets are shown in **Fig 3**. From the graph, it can be observed that the drugs with high binding energies were found to have maximum therapeutic indications for CNS (central nervous system) followed by immunological disorder with the least for pain management. Whereas, the drugs with low binding energies were mostly found to have therapeutic indications for immunological disorders followed by drugs for vitamin deficiencies and anesthetics.

## Identification of existing approved drugs for COVID-19

According to the WHO report (WHO-2019-nCoV-therapeutics-2022) two known drugs namely remdesivir (antiviral compound) and baricitinib (anti-arthritis compound) are reported to be approved for the treatment of COVID-19. Along with these, several studies have reported a list of drugs under clinical trials and those used for emergency treatment of COVID-19. In the current study, these known drugs were identified among the drugs interacting with 24 SARS-CoV2 targets. Although the existing FDA approved drugs and drugs which are under clinical trials for COVID-19 were not found among the top 10 drugs, few of them showed good binding affinity towards the selected SARS-CoV2 targets and have been highlighted along with the top 10 drugs as shown in **S3a, S3c Fig in S1 File**. From the graphs it can be observed that the known drugs remdesivir, baricitinib, ritonavir, hydroxychloroquine, lopinavir, favipiravir and dexamethasone had good docking scores due to stable interaction with almost all the 24 SARS-CoV2 targets. The two reported approved drugs for COVID-19,

**Table 3. List of potential top 10 FDA approved drugs for repurposing against structural SARS-CoV2 targets along with their known therapeutic indication.**

| Protein | Drug Name | Therapeutic Indication |
|---|---|---|
| Envelope | Abiraterone Acetate | Liver disease |
| | Flufenamic Acid | Pain |
| | Gliclazide | Diabetes mellitus type 2 |
| | Palonosetron | Nausea and vomiting |
| | Trilostane | Hypercortisolism |
| | Florantyrone | Dyskinesia |
| | Niclosamide | Helminthiasis |
| | Perampanel | Partial seizure |
| | Triazolam | Insomnia |
| | Bentiromide | Pancreas Function |
| Membrane | Nilotinib | Anticancer |
| | Ergotamine | Migraine |
| | Tirilazad | CNS disorder |
| | Dihydroergotamine | Migraine |
| | Antrafenine | Inflammation |
| | Zafirlukast | Asthma |
| | Midostaurin | Leukemia |
| | Acetic Acid | Susceptible infections |
| | Conivaptan | Liver disease |
| | Pranlukast | Asthma |
| Spike | Talazoparib | Anticancer |
| | Nilotinib | Anticancer |
| | Prednisolone | Antiinflamatory |
| | Ruboxistaurin | Diabetes Mellitus |
| | Etravirine | HIV-1 infection |
| | Piketoprofen | Pain |
| | Danazol | Endometriosis |
| | Lumacaftor | Cystic fibrosis |
| | Raltegravir | HIV infection |
| | Revaprazan | Gastric ulcer |
| Nucleoprotein | Paritaprevir | Chronic hepatitis C |
| | Dihydroergotamine | Migraine |
| | Temoporfin | Anticancer |
| | Suramin | African trypanosomiasis |
| | Ergotamine | Migraine |
| | Lurbinectedin | Anticancer |
| | Ledipasvir | Antiviral |
| | Acetyldigoxin | Cardiovascular |
| | Conivaptan | Liver disease |
| | Lanatoside C | Cardiovascular |

remdesivir and baricitinib were found to be interacting with all the SARS-CoV2 targets except for envelope and ORF7a proteins. Several studies have reported remdesivir as an antiviral drug to inhibit the function of RNA-dependent RNA polymerase (NSP12) with a binding score of -5.0 kcal/mol to -10 kcal/mol [52–54]. From the study, it can be compared that remdesivir has a binding score of -7.0 kcal/mol with NSP12 which confirms the significance of the study. The highest docking score of -8.1 kcal/mol was observed with membrane protein, while the lowest

**Table 4. List of potential top 10 FDA approved drugs for repurposing against non-structural SARS-CoV2 targets along with their known therapeutic indication.**

| Protein | Drug Name | Therapeutic Indication |
|---|---|---|
| NSP1 | Thebacon | Pain |
| | Carbamazepine | Bipolar disorder |
| | Benmoxin | Depression |
| | Piketoprofen | Pain |
| | Imatinib | Leukemia |
| | Methysergide | Migraine |
| | Diacerein | Pain |
| | Levosimendan | Cardiovascular |
| | Plafibride | Antimetabolites |
| | Prenoxdiazine | Cough suppressants |
| NSP2 | Suramin | African trypanosomiasis |
| | Teicoplanin Aglycone | Antibacterial |
| | Ergotamine | Migraine |
| | Dihydroergotamine | Migraine |
| | Glecaprevir | Chronic hepatitis C |
| | Zorubicin | None |
| | Paritaprevir | Chronic hepatitis C |
| | Dihydroergocornine | Neurotransmitter Agents |
| | Cetrorelix | Hormone Antagonists |
| | Midostaurin | Immunomodulating agents |
| NSP3 | Scopolamine Butylbromide | Gastric spasm |
| | Pazopanib | Anticancer |
| | Venetoclax | Anticancer |
| | Naldemedine | Schizophrenia |
| | Nafamostat | Cystic fibrosis |
| | Zafirlukast | Asthma |
| | Bictegravir | HIV infection |
| | Bagrosin | Anti-epileptic |
| | Ergotamine | Migraine |
| | Rebamipide | Anticancer |
| NSP4 | Radotinib | Anticancer |
| | Talniflumate | Cystic fibrosis |
| | Netarsudil | Ocular hypertension |
| | Exatecan | Antineoplastic Agents |
| | Picloxydine | Infection |
| | Trenbolone | Anabolic Agents |
| | Vibegron | Muscle dysfunction |
| | Pranlukast | Asthma |
| | Flibanserin | Sexual disorder |
| | Tasosartan | Cardiovascular |
| NSP5 | Siponimod | Relapsing multiple sclerosis |
| | Progesterone | Contraceptive |
| | Cefazolin | Bacterial infecton |
| | Suramin | African trypanosomiasis |
| | Tasosartan | Antihypertensive |
| | Midostaurin | Leukemia |
| | Talazoparib | Anticancer |
| | Exatecan | Antineoplastic agents |
| | Casopitant | Urinary incontinence |
| | Dactinomycin | Sarcoma |

*(Continued)*

**Table 4.** (Continued)

| Protein | Drug Name | Therapeutic Indication |
|---|---|---|
| NSP6 | Lomitapide | Hypercholesterolemia - |
| | Moxidectin | Infection |
| | Diphenadione | Antithrombotic agents |
| | Avapritinib | Gastrointestinal stromal tumor |
| | Temoporfin | Anticancer |
| | Vibegron | Muscle dysfunction |
| | Tadalafil | Benign prostatic hyperplasia |
| | Phenolsulfonphthalein | Renal function study |
| | Bisdequalinium Chloride | Antibacterial |
| | Ketotifen | Allergic conjunctivitis |
| NSP7 | Desloratadine | Allergy |
| | Mianserin | Depressive disorder |
| | Setiptiline | Depressive disorder |
| | Mecloqualone | Insomnia |
| | Metapramine | Antidepressive Agents |
| | Cyproheptadine | Allergic symptoms |
| | Flufenamic Acid | Pain |
| | Methaqualone | Antidepressive |
| | Oxcarbazepine | Depressive disorder |
| | Lorajmine | Cardiovascular |
| NSP8 | Bisdequalinium Chloride | Antibacterial |
| | Alpelisib | Cancer |
| | Niraparib | Tumor of ovary |
| | Risdiplam | Spinal muscular atrophy |
| | Fenoverine | Irritable bowel syndrome. |
| | Irinotecan | Anticancer |
| | Calcipotriol | Psoriasis |
| | Dihydroergotamine | Migraine |
| | Florantyrone | Dyskinesia |
| | Paliperidone | Schizophrenia |
| NSP9 | Temoporfin | Anticancer |
| | Irinotecan | Cancer |
| | Zorubicin | Anticancer |
| | Ciclesonide | Allergic rhinitis |
| | Risdiplam | Spinal muscular atrophy |
| | Ledipasvir | Antiviral |
| | Suramin | African trypanosomiasis |
| | Venetoclax | Anticancer |
| | Talniflumate | Anti-inflammatory |
| | Quinbolone | Hormonal |
| NSP10 | Dihydroergocristine | Progressive mental decline |
| | Ergometrine | Postpartum haemorrhage |
| | Ubidecarenone | Congestive heart failure |
| | Tacrine | CNS disorder |
| | Perflubron | Hemorrhage |
| | Rasagiline | Parkinson's disease |
| | Talastine | Antihistamine |
| | Barnidipine | Hypertension |
| | Glasdegib | Acute myeloid leukemia |
| | Lercanidipine | Hypertensive disorder |

*(Continued)*

**Table 4.** (Continued)

| Protein | Drug Name | Therapeutic Indication |
|---|---|---|
| NSP12 | Dihydroergotamine | Migraine |
| | Ergotamine | Migraine |
| | Trabectedin | Tumor of ovary |
| | Carsalam | Anti-inflammatory |
| | Mosapramine | Schizophrenia |
| | Amrubicin | Lung cancer |
| | Betamethasone Phosphate | Anti-inflamation |
| | Rimegepant | Migraine |
| | Midostaurin | Leukemia |
| | Pirarubicin | Immunomodulating agents |
| NSP13 | Pazopanib | Anticancer |
| | Venetoclax | Anticancer |
| | Racepinefrine | Hypertensive disorder |
| | Zafirlukast | Asthma |
| | Bictegravir | HIV infection |
| | Prednisolone Succinate | Anaphylaxis, asthma |
| | Ergotamine | Migraine |
| | Rebamipide | Anticancer |
| | Tucatinib | Anticancer |
| | Steviolbioside | Antitubercular agent |
| NSP14 | Rimegepant | Migraine |
| | Tucatinib | Anticancer |
| | Midostaurin | Leukemia |
| | Radotinib | Anticancer |
| | Fosaprepitant | Nausea and vomiting |
| | Alatrofloxacin | Bacterial infections |
| | Lumacaftor | Cystic fibrosis |
| | Lonafarnib | Hutchinson-Gilford syndrome |
| | Nicomorphine | Analgesic |
| | Tacalcitol | Plaque psoriasis |
| NSP15 | Lumacaftor | Cystic fibrosis |
| | Fazadinium | Pain |
| | Evocalcet | Secondary hyperparathyroidism |
| | Ivosidenib | Acute myeloid leukemia |
| | Tasosartan | Antihypertensive |
| | Fluspirilene | Schizophrenia |
| | Olodaterol | Lung disease |
| | Argatroban | Thrombosis |
| | Hesperidin | Hemorrhoids |
| | Quercetin | Kidney disease |
| NSP16 | Sonidegib | Basal cell carcinoma of skin |
| | Picloxydine | Infection |
| | Flavin Adenine Dinucleotide | Vitamin B2 deficiency |
| | Gitoxin | Anticancer |
| | Ceftobiprole Medocaril | Pneumonia |
| | Zorubicin | Anticancer |
| | Naldemedine | Schizophrenia |
| | Enantate Benzilic Acid | Hormonal deficiency |
| | Tirilazad | CNS disorder |
| | Berotralstat | Angioneurotic edema |

**Table 5. List of potential top 10 FDA approved drugs for repurposing against accessory SARS-CoV2 targets along with their known therapeutic indication.**

| Protein | Drug Name | Therapeutic Indication |
|---|---|---|
| ORF3a | Bisdequalinium Chloride | Antibacterial |
| | Dutasteride | Benign prostatic hyperplasia |
| | Temoporfin | Anticancer |
| | Tirilazad | CNS disorder |
| | Ergotamine | Migraine |
| | Dihydroergotamine | Migraine |
| | Teicoplanin Aglycone | Antibacterial |
| | Nilotinib | Anticancer |
| | Revefenacin | Chronic obstructive lung disease |
| | Adapalene | Dermatology |
| ORF6 | Bisdequalinium Chloride | Antibacterial |
| | Tirilazad | CNS disorder |
| | Temoporfin | Anticancer |
| | Vaniprevir | Chronic hepatitis C |
| | Adapalene | Dermatology |
| | Saquinavir | HIV infection |
| | Conivaptan | Liver disease |
| | Dihydroergotamine | Migraine |
| | Fluspirilene | Schizophrenia |
| | Paritaprevir | Chronic hepatitis C |
| ORF7a | Hydroflumethiazide | Hypertensive disorder |
| | Naftazone | Hemorrhoids |
| | Pomalidomide | Multiple myeloma |
| | Tramazoline | Nasal decongestion |
| | Dezocine | Pain |
| | Huperzine A | Alzheimer's disease |
| | Hydromorphone | Pain |
| | Naproxen | Pain |
| | Phenformin | Type II diabetes mellitus |
| | Primidone | Epilepsy |
| ORF 8 | Tirilazad | CNS disorder |
| | Pirenoxine | Cataract |
| | Vesnarinone | Antiviral agents |
| | Magnesium Orotate | Mineral supplements |
| | Tropatepine | CNS disorder |
| | Thebacon | Respiratory system |
| | Enoxolone | Anti-inflammatory agents |
| | Venetoclax | Anticancer |
| | Traxanox | Immunosuppressive Agents |
| | Zorubicin | Immunomodulation agent |
| ORF 9b | Paritaprevir | Chronic hepatitis C |
| | Ergotamine | Migraine |
| | Bisdequalinium Chloride | Antibacterial |
| | Ubrogepant | Migraine |
| | Dihydroergotamine | Migraine |
| | Mosapramine | Schizophrenia |
| | Venetoclax | Anticancer |
| | Naldemedine | Schizophrenia |
| | Netupitant | Nausea and vomiting |
| | Telmisartan | Hypertensive disorder |

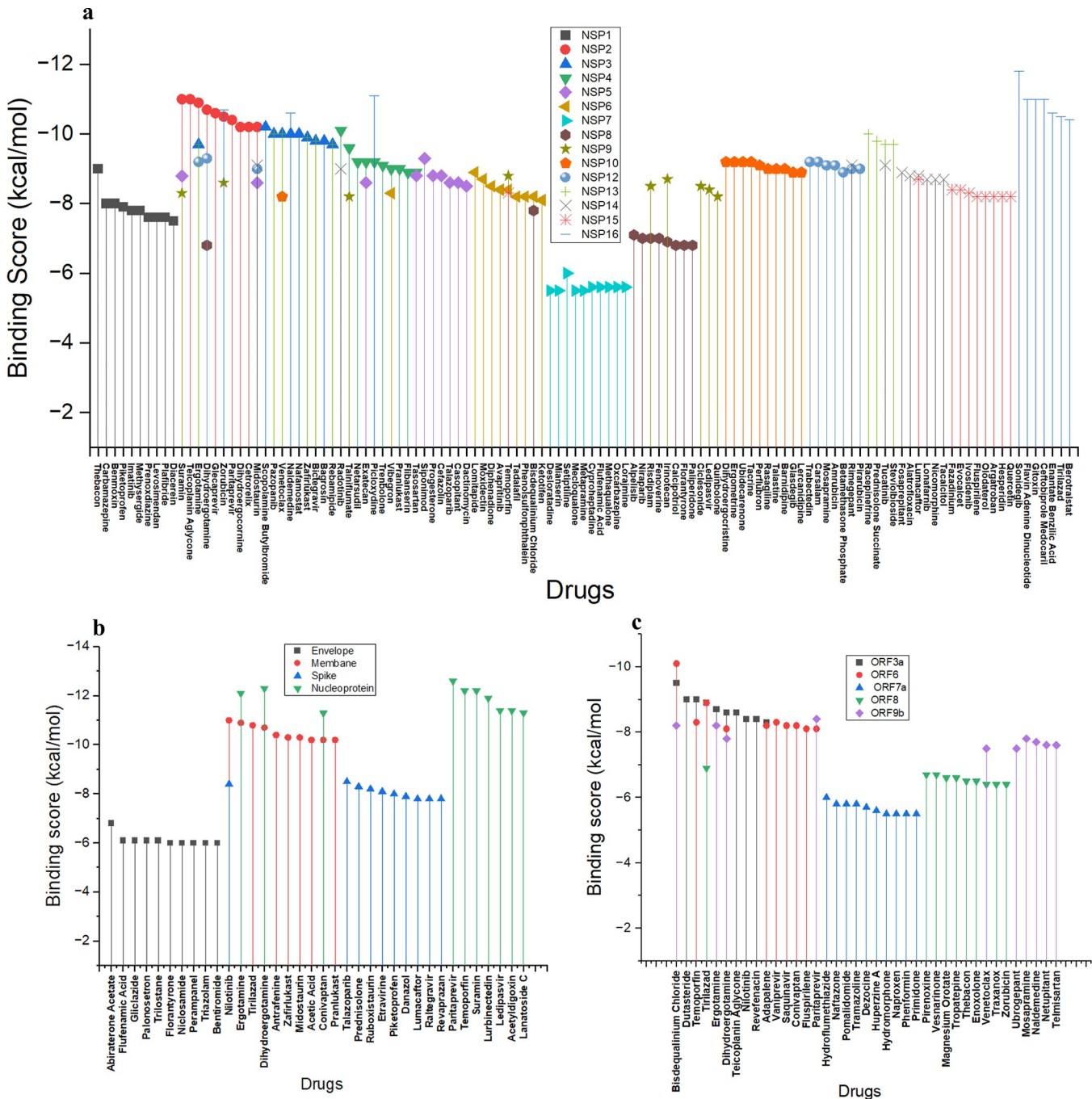

**Fig 2. The binding score distribution of top 10 FDA approved drugs against SARS-CoV2 proteins.** The figure depicts the distribution of top the 10 FDA approved drugs against SARS-CoV2 proteins namely, a) non-structural proteins namely NSP1, NSP2, NSP3, NSP4, NSP5, NSP6, NSP7, NSP8, NSP 9, NSP10, NSP12, NSP13, NSP14, NSP15 and NSP16, b) structural proteins namely Envelope, Spike and Nucleoprotein, and c) accessory proteins namely ORF3a, ORF6, ORF7a, ORF8, ORF9.

docking score of -2.2 kcal/mol with NSP7 and an average binding score of -6.5 kcal/mol across the interacting targets. Similarly, baricitinib an anti-arthritis drug, is reported to interrupt the entry and intracellular assembly of SARS-CoV2 into the host cell and interfere with the immune response [55–58]. In this regard, baricitinib was observed to have the highest docking

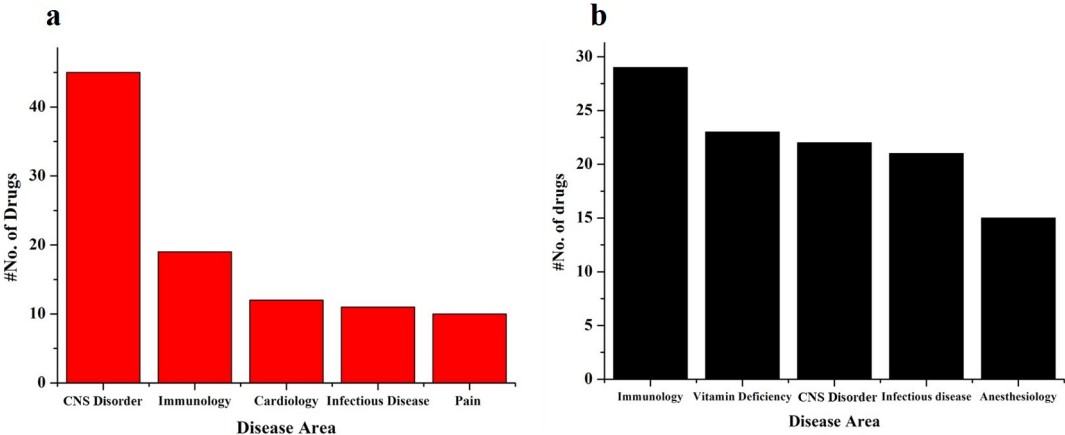

**Fig 3.** The classification of a) top 10 and b) bottom 10 drugs based on their therapeutic areas. The figure depicts that the drugs with good docking score are mainly used for neurological disorders whereas poor docking score drugs are mainly from immune related disease.

score of -7.9 kcal/mol with membrane, which is involved in humoral response during the SARS-CoV2 and host interaction. Meanwhile, the lowest score of -3.4 kcal/mol was observed for nucleoprotein with an average binding score of -5.9 kcal/mol across the interacting targets, where some of the targets are involved in viral entry and immune response. This suggests that some of the existing known drugs for COVID-19 have shown good binding energy along with the top 10 drugs, which can be of significant importance for repurposing against multiple SARS-CoV2 targets.

## Privileged scaffold identification and their analysis

For the identification of the privileged scaffolds, a total of 162 unique drugs present in the top 10 drugs across the 24 SARS-CoV2 targets were taken to generate 3 levels of scaffolds. The scaffold tree displayed three levels 1,2, and 3, which consist of 71, 103 and 123 scaffolds respectively. The identified scaffolds were then compared with the scaffolds generated in each level for the existing FDA approved drugs and drugs under clinical trials for COVID-19 disease. From the analysis, four common scaffolds (6,7,8,9-Tetrahydro-5H-cyclohepta[c]pyridine, 1-Benzazepine, decalin and leucoline) in level 2 and two common scaffolds (4,5,6,7,8,8a,9,10-Octahydro-2(3H)-phenanthrenone, Gona-1,3,5(10)-trien-3-ol) in level 3 of the scaffold tree were observed and are shown in **Fig 4**. The compounds present in level 2 and 3 of the scaffold trees generated from the unique compounds can be considered as potential candidates for designing drugs to inhibit the effect of SARS-CoV2. In addition, 2D interaction analysis was carried out to analyse the interactions of the identified scaffold with 24 SARS-CoV2 targets. Targets namely, envelope, membrane, NSP4, NSP5, NSP14 and ORF9b were observed to have interactions with the scaffolds forming bonds such as π-alkyl, π - π stacked, conventional hydrogen bonds, π -sigma and mostly alkyl bonds **(S4 Table in S1 File)**. The common scaffolds along with the interacting site information can also serve as starting points for the development of novel SARS-CoV2 inhibitors.

## Polypharmacological property analysis

The effectivity of antivirus progressively goes down due to the increased rate of viral mutation and develops resistance toward the drug action [59]. Hence, targeting multiple targets involved in the same or different pathways through polypharmacology approach [60, 61] seems to be an

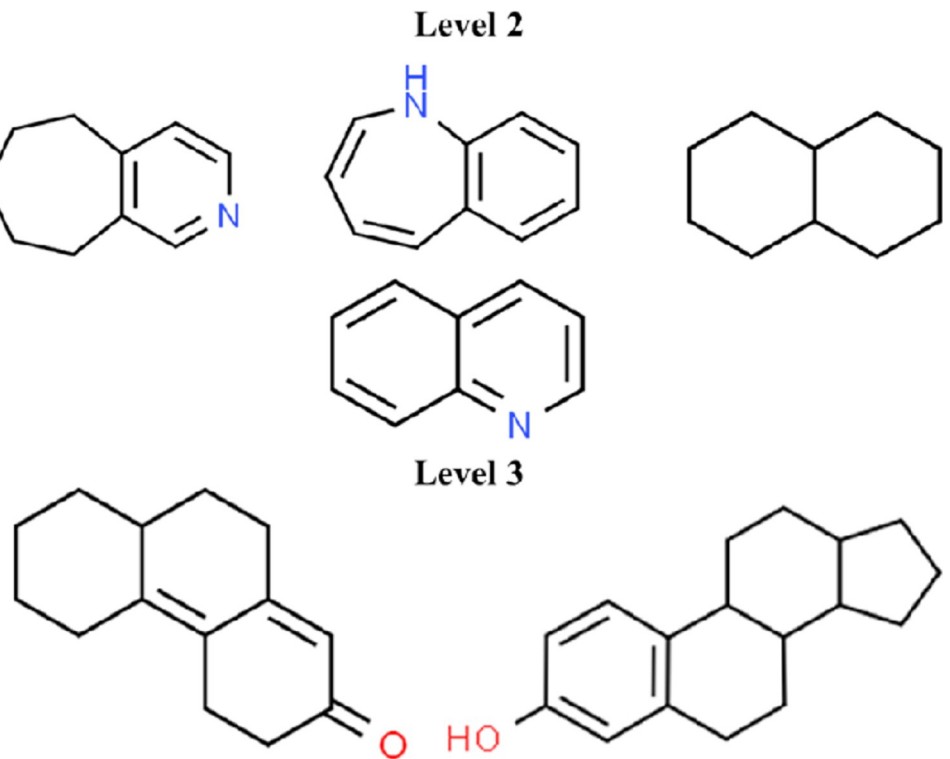

**Fig 4. Scaffold analysis of top scored drugs across 24 SARS-CoV2 targets.** The level 2 and level 3 common scaffolds that are present among the top scored drug molecules across all the 24 SARS-CoV2 targets, existing approved drugs and drugs under clinical trials for COVID-19.

effective strategy for combating a complex disease like COVID-19. In this regard, the FDA approved drugs in the top ten list were filtered based on the drugs interacting with multiple targets of more than 3 SARS-CoV2 proteins. A graph was generated representing the number of interacting targets against the drug as shown in **Fig 5**.

From **Fig 5**, it can be observed that out of the total number of 162 unique drugs present in the top 10 drugs across the 24 SARS-CoV2, 15 drugs were found to have interaction with 3 or more targets. Among these drugs, dihydroergotamine and ergotamine showed a maximum number of interactions with around 8 SARS-CoV2 targets. Drugs such as bisdequalinium chloride, midostaurin, temoporfin, tirilazad and venetoclax had interaction with around 5 numbers of SARS-CoV2 targets. These drugs have therapeutic indications for migraine, bacterial infection, central nervous system disorder and most commonly cancer (**Table 6**). Several studies have reported that dihydroergotamine and ergotamine are having stable interactions with 3CL$^{pro}$ and also it has the polpypharmacology efficacy [55, 62]. In addition, the drugs tirilazad and venetoclax were reported to show stable binding affinity towards multiple targets for SARS-CoV2 [35, 56], revealing their polypharamocological property.

## Protein-ligand interaction analysis

One of the most important steps in the drug development process is understanding how drugs interact with target proteins and identifying the important interacting residues [63]. Thus, the 15 drugs interacting with more than 3 SARS-CoV2 targets were further taken for 2D interaction analysis with their respective proteins using Discovery studio. This was carried out to identify the important interacting residues along with the non-covalent bonds contributing to

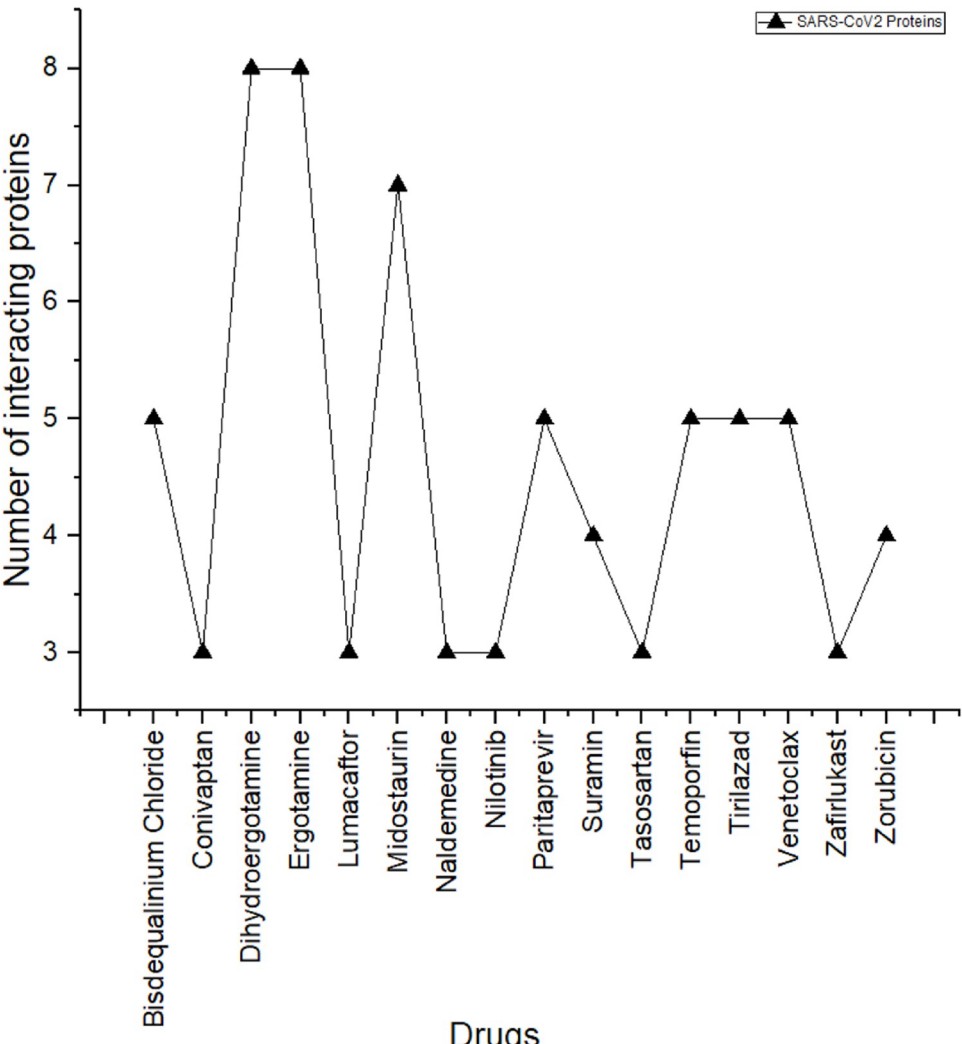

**Fig 5. Representation of drugs interacting with multiple SARS-CoV2 targets.** A total of 15 drugs are observed to have interaction with multiple targets (more than 3) and the maximum number of targets are found to be interacting with Dhydroergotamine, Ergotamine and Midostaurin.

the protein-ligand binding. Among the multi-targeting drugs, dihydroegotamine and ergotamine were observed to have interaction with the maximum number (8) of SARS-CoV2 proteins which is shown in **Fig 6**. It can be observed that both the drugs have interaction with eight proteins and shares common interacting proteins such as membrane, NSP2, NSP12, nucleoprotein, ORF3a and ORF9b protein. Further, the detailed 2D interactions of drugs with eight, five, four and three numbers of SARS-CoV2 targets are given in **S4a-S4d Fig in S1 File**. From the figures depicting the 2D interactions of drugs and proteins, it can be observed that amino acid residues such as ILE, LEU, ALA, ARG and VAL are some of the most commonly occurring amino acids involved in the overall interactions. These amino acids are mostly found to be hydrophobic in nature and hydrophobic interactions are reported to be the driving force between protein-ligand interactions as it contributes to the binding affinity of the complexes [64]. Among the various non-covalent bonds occurring in a drug-protein interaction, hydrogen bond plays a significant role in forming the protein-ligand complex and these interactions are reported to be important for consideration while designing inhibitors against a

**Table 6. List of FDA approved drugs interacting with more than three SARS-CoV2 targets along with their original indications.**

| Sl.no | Drug | Original Indication | Targets |
|---|---|---|---|
| 1. | Bisdequalinium Chloride | Bacterial infection | NSP6, NSP8, ORF3a, ORF6, ORF9b |
| 2. | Conivaptan | Liver disease | Membrane, Nucleoprotein, ORF6 |
| 3. | Dihydroergotamine | Migraine | Membrane, Nucleoprotein, NSP2, NSP8, NSP12, ORF3a, ORF6, ORF9b |
| 4. | Ergotamine | Migraine | Membrane, Nucleoprotein, NSP2, NSP3, NSP12, NSP13, ORF3a, ORF9b |
| 5. | Lumacaftor | Cystic fibrosis | Spike, NSP14, NSP15 |
| 6. | Midostaurin | Cancer | Membrane, NSP2, NSP5, NSP12, NSP14 |
| 7. | Naldemedine | Hemorrhoids | NSP3, NSP16, ORF9b |
| 8. | Nilotinib | Analgesic | Membrane, Spike, ORF3a |
| 9. | Paritaprevir | Nausea and vomiting | Nucleoprotein, NSP2, ORF6, ORF9b |
| 10. | Suramin | African trypanosomiasis | Nucleoprotein, NSP2, NSP5, NSP9 |
| 11. | Tasosartan | Cardiovascular disease | NSP4, NSP5, NSP15 |
| 12. | Temoporfin | Cancer | Nucleoprotein, NSP6, NSP9, ORF3a, ORF6 |
| 13. | Tirilazad | Central nervous system | Membrane, NSP16, ORF3a, ORF6, ORF8 |
| 14. | Venetoclax | Cancer | NSP3, NSP9, NSP13, ORF8, ORF9b |
| 15. | Zafirlukast | Asthma | Membrane, NSP3, NSP13 |
| 16. | Zorubicin | Cancer | NSP2, NSP9, NSP16, ORF8 |

target [65]. In regard to this, it can be observed that polar-amino acids [66] such as SER, ASN, GLN, THR are involved majorly in hydrogen formation (**S5 Table in S1 File**). Other positively charged and non-polar amino acids like ARG and GLY respectively are also found to be involved in the hydrogen bond formation. Along with hydrogen bonds, other non-covalent interactions such as π-alkyl, π - π stacked, conventional hydrogen bonds, π -sigma and alkyl bonds are observed in the drug-protein interaction. The hydrophobic and hydrogen bond forming residues are also observed among the active site residues of the 24 SARS-CoV2 proteins as shown in **S1a, S1b Fig in S1 File**. For example, the active site residue of NSP12 was found to interact with Dihydroergotamine through the residue **ARG553**, which is involved in the formation of hydrogen bonds. In the case of ergotamine, the interacting residues and active site residues of the membrane were **THR52**, and for NSP3, it was **GLY163** and **GLU167**, which are involved in -sigma bonds and hydrogen bonds, respectively. Similarly, among the drugs interacting with 5, 4 and 3 targets it was observed that few of the active site residues of the protein were shown to be among the interacting residues which are hydrophobic in nature and are mainly involved in hydrogen bond formation. These identified residues involved in the formation of drug-protein interactions will help in understanding the therapeutic effect of a drug against the target and also in identifying the important residues which can be targeted while designing high binding and target specific inhibitors.

## Pathway analysis for drug repurposing

**Gene enrichment analysis.** Analyzing the biological pathway of a particular drug's targets can provide valuable insight into the mechanism of a drug's action and its impact on repurposing the drug for new indications [67]. 15 drugs interacting with multiple targets were selected and their original targets were obtained from publicly accessible databases such as Drug bank [44] and Drug Central [45]. A total of 26 targets have been obtained, and a gene enrichment analysis was performed using the network analysis tool ShinyGo (http://bioinformatics. sdstate.edu/go/). The gene enrichment analysis was performed for the genes involved in biological processes, cellular components, and molecular function pathways. A cut-off minimum 0.05 P-value was maintained for FDR (false discovery rate) and the top 20 pathways were

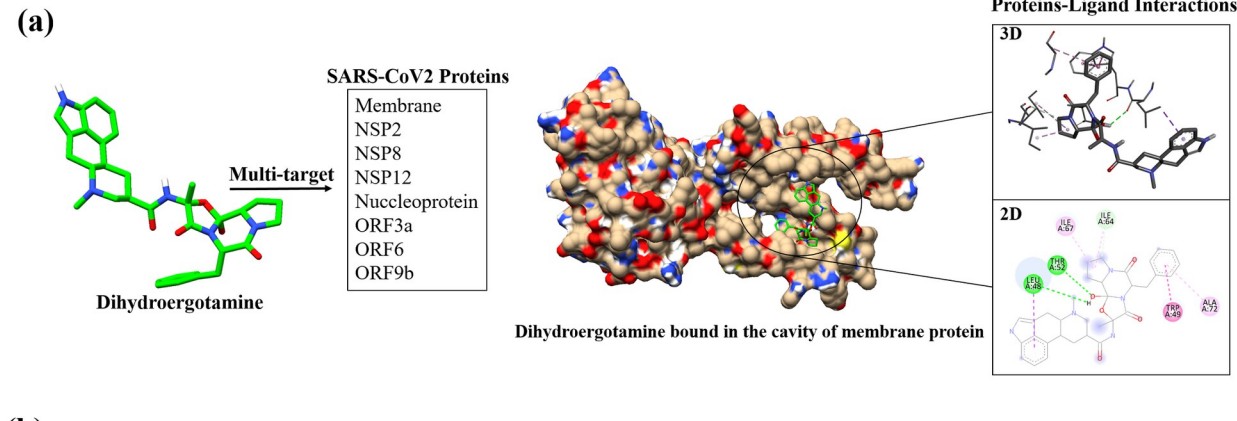

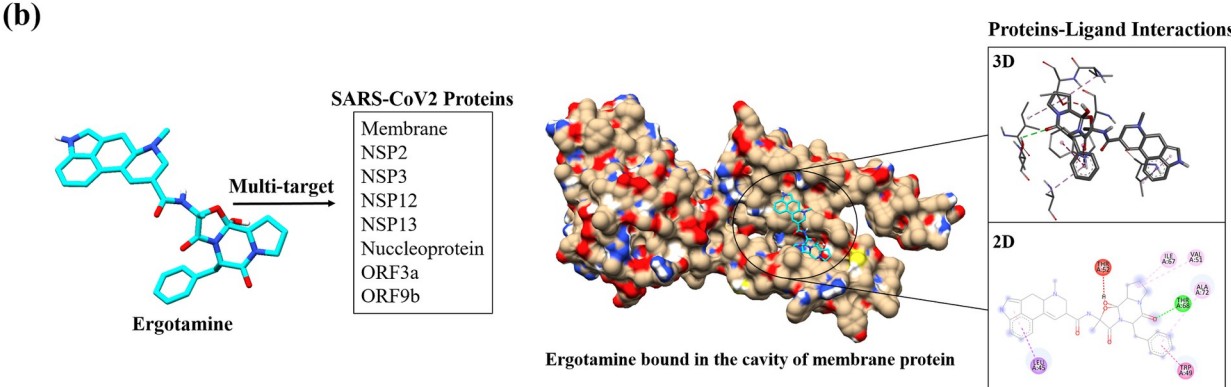

**Fig 6. Representation of 2D interaction of drugs interacting with 8 SARS-CoV2 proteins.** Among the drugs interacting with multiple targets, (a) Dihydroergotamine and (b) ergotamine was observed to have interaction with the maximum number of SARS-CoV2 targets (8 targets). The figure represents an example of 2D interaction with membrane protein among the 8 proteins commonly for both the drugs. The drug molecule is seen to bound in the cavity of the protein through various non-covalent interactions with the amino acid residues that contributes to the protein-ligand binding affinity.

selected for further analysis. **Fig 7** depicts the constructed network and gene characteristic plot.

The network in **Fig 7** depicts the interacting nodes of genes involved in biological processes, cellular components, and molecular function respectively. It displays the functional categories of genes that are most highly enriched as well as how the pathways have connected the genes. From the enrichment plot, it can be observed that the majority of genes in the biological process categories were involved in the control of biological quality. This indicates that most of the genes in the biological process are found to be involved in the regulation and control of the biological quality of cellular functions such as cell cycle regulation, DNA repair, apoptosis etc [68]. Due to the short size of their genomes, viruses have a restricted ability to code and as a result, they use the host cellular mechanism and components to enable them to replicate [69]. In the process, the virus influences the host cell cycle leading to failure in safeguarding the host cell thus affecting host DNA replication and repair mechanism [69]. The pathway with the most preferable FDR score was observed to be phospholipase C-activating G protein-coupled receptor (GPCR) signalling pathway (**Fig 7A**). GPCR signalling pathway plays an important role in governing biological events such as cell differentiation, apoptosis and also cellular growth and is known to interact with integral membrane proteins [70]. Studies have indicated the connection between the host immune response during SARS-CoV2 infections and the G protein-coupled receptor signalling pathway, as the virus enters the host by attaching to a

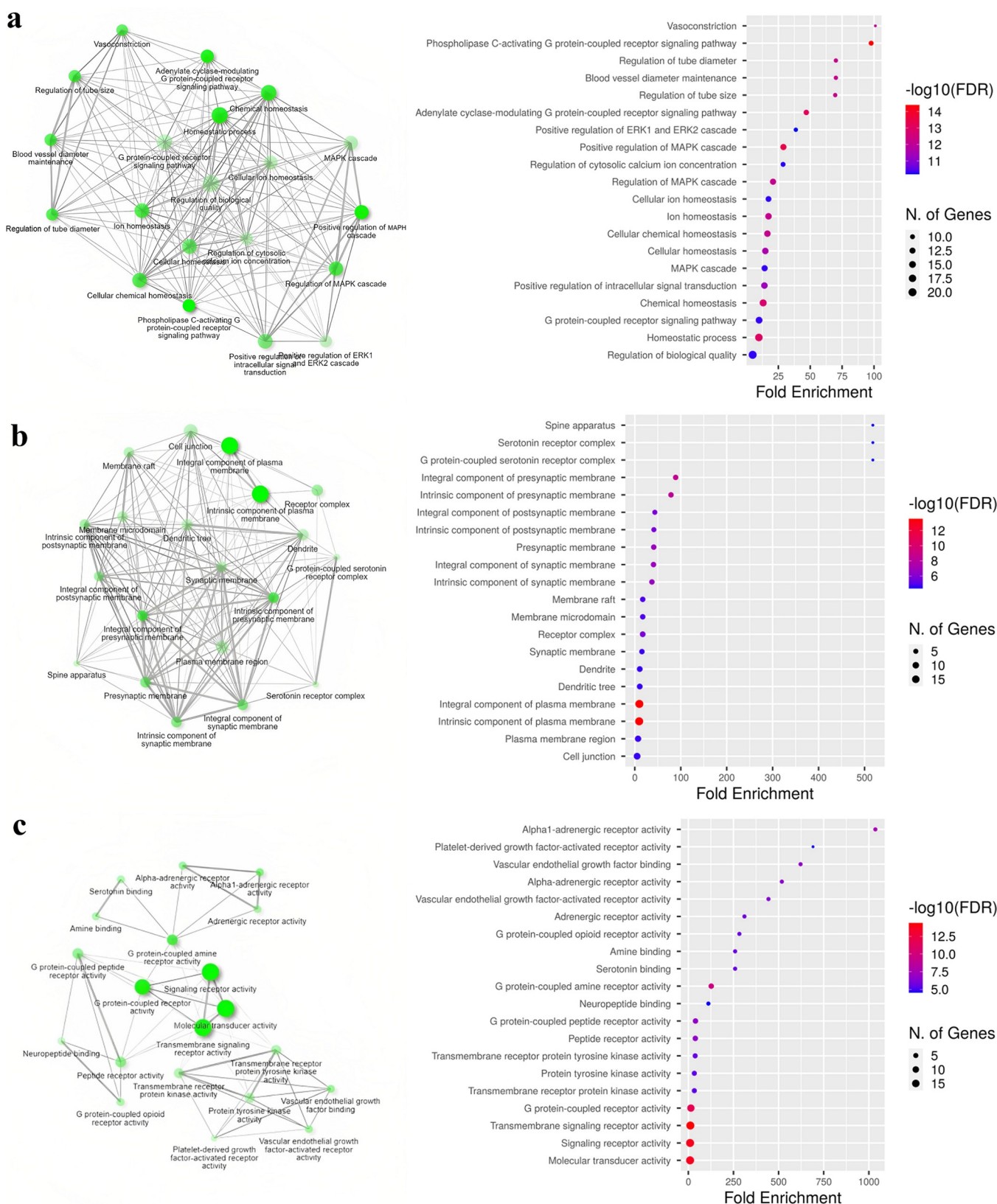

**Fig 7.** Gene enrichment analysis for a) Biological process, b) Cellular component and c) Molecular function of genes associated with top screened drugs interacting with multiple SARS-CoV2 targets. The network represents the interaction of nodes (genes involved in biological processes, cellular components and molecular functions) connected through edges. From the enrichment plot, it can be observed that for biological processes, the maximum number of gene is involved in regulation of biological quality, for Cellular component, the maximum number of genes are involved in cell junction and for Molecular function, the maximum number of genes are involved in molecular transducer activity.

GPCR namely angiotensin-converting enzyme 2 (ACE2) receptor [71]. In the category of cellular components, the majority of the genes were found to be a part of the integral component of plasma membrane and this pathway was also observed to have the most preferable FDR score (**Fig 7B**). The plasma membrane is a highly specific permeable barrier that encompasses cell components and performs a number of functions such as sustaining the cell structure and shape, controlling the transfer of stimulants signal transmission [72]. According to a study reported by Deng *et al.* (2021), the human transmembrane protease serine 2 is activated when the SARS-CoV2 enters the host cell through endocytosis and directs fusing of the viral membrane with the host plasma membrane [73]. So, it is probable that the genes identified as a part of the integral component of the plasma membrane are essential for preserving cell functionality and the proper administration of membrane during SARS-CoV2 infection. Further, in the molecular function category, the maximum number of genes were involved in molecular transducer activity (**Fig 7C**). This includes genes that encode receptors that are involved in ion channelling and cell signalling [72]. According to several studies, the number of certain key ions present in channelling and signalling during the viral-host attachment is an important factor that influences the attachment, infection and invasion of SARS-CoV2 in the host [74]. In addition, from the enrichment plot, it was observed that the pathway associated with G protein-coupled receptor activity showed the lowest FDR score similar to the pathway in the biological process as mentioned earlier. This indicates the significance of genes involved in this pathway and the crucial role of G protein-coupled receptor activity in the immune response during the SARS-CoV2 infection [70, 71]. Thus, identifying the involvement of drugs' genes in pathways linked to various biological processes during SARS-CoV2 infection supports the finding of the current study in determining potential drugs that may be repurposed against SARS-CoV2. From the overall analysis, it can be seen that majority of the top 10 drugs are predominantly involved in pathways related to fundamental cellular mechanisms like cell cycle regulation, cell functionality, cell signalling, transmission, ion channelling, DNA repair, apoptosis and immune response (**S6 Table in S1 File**) This suggests that these pathways and the genes involved in the pathways are significant targets for drug repurposing and may have therapeutic application for COVID-19 disease.

**KEGG pathway analysis.** KEGG pathway analysis was carried out to study the functional role of genes and identify specific pathways that are differentially regulated depending on the disease condition or drug being used. The 26 genes linked to the 15 drugs were taken for the KEGG pathway analysis using the online server Metascape (https://metascape.org) and the parameters for calculation was maintained as default [75]. From the study, enriched ontology clusters, a protein-protein interaction network and an enrichment heatmap was generated (**Fig 8**) which were analysed for identifying particular biochemical interactions and response networks involved in a biological process.

In **Fig 8**, the enriched gene cluster (**Fig 8C**) represent clusters of genes in colours which are involved in a particular pathway and size of the cluster represents the number of genes involved. Each cluster is connected by edges and the thickness of the edges indicates the level of similarity among the clusters of pathways. From the enriched gene cluster it can be observed that majority of the genes are involved in specific role such as G protein-coupled receptor activity, positive regulation of mitogen-activated protein kinase (MAPK) cascade, signalling

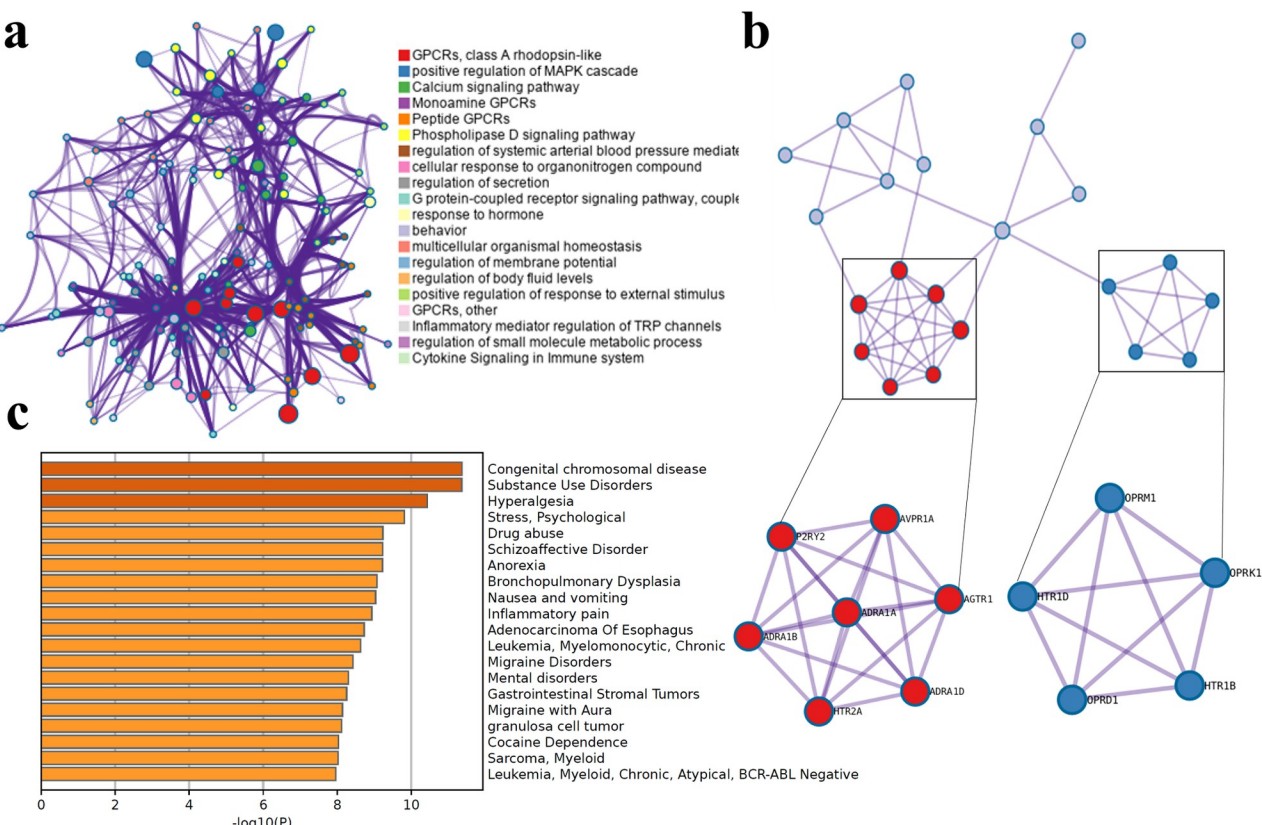

**Fig 8. Pathway analysis of genes related to the 15 drugs interacting with multiple proteins of SARS-CoV2 was carried out using online server Metascape.** a) Enriched ontology clusters represent the clusters of genes which are involved in the different pathway and the nodes of the colour represent the genes belonging to the same clusters. The nodes are linked by edges where the thickness of the edges represents the similarity score of the genes. b) Protein-protein interaction network of the genes involved in different pathways. The proteins which are most densely connected have been clustered through MCODE algorithm to identify the protein neighbourhoods. c) Enrichment analysis of genes-disease association represents the genes which are involved in different diseases. The images were generated using Metascape [71].

pathways, cellular response and regulation. These pathways involved in G protein-coupled receptor activity, cellular response and regulation and cell signalling have also been identified in the gene enrichment analysis and linked to SARS-CoV2 infection [70–72]. From the protein-protein interaction network (**Fig 8B**), it can be observed that two main clusters of protein that are densely connected to each other have been identified among the various other clusters of proteins obtained through Molecular Complex Detection (MCODE). The densely connected proteins represents highly interconnected clusters which are associated to a particular biological process and can be analysed to identify potential targets for therapeutics [76]. From the identified clusters, it has been observed that 12 genes (ADRA1A, HTR2A, ADRA1D, P2RY2, ADRA1B, AVPR1A, AGTR1, OPRM1, HTR1B, OPRD1, OPRK1, HTR1D) are highly interconnected. These genes were mainly found to be involved in pathways such as G protein-coupled receptor activity, cellular response and regulation, cell signalling, immune response and apoptosis in both the gene enrichment and KEGG pathway analysis. The presence of these genes and pathways in both the analysis indicates the important role of these genes and the pathways involved in various biological processes particularly in SARS-CoV2 infection. In addition, the enrichment analysis of genes-disease association (**Fig 8C**) has shown the involvement of genes mostly in diseases related to neurological disorder, pain and some in cancer.

This supports the identification of multi targeting drugs among the top 10 drugs with therapeutic indication for migraine, bacterial infection, neurological disorder and cancer.

## Discussion

The study involved screening of FDA approved drugs against the 24 SARS-CoV2 targets, followed by drug categorization into most active and least active based on high and low docking score respectively across the targets. The docking scores of the top 10 FDA approved drugs against the 24 SARS-CoV2 targets obtained from Autodock Vina were validated using MT-DTI, SwissDock, and iGEMDOCK and it was observed that similar docking scores were obtained through all the three methods. Upon categorization, the therapeutic indications of the drugs were allocated to identify the categories of drugs with significant therapeutic indications that may have the potential for repurposing against SARS-CoV2. It was found that the majority of the compounds with high docking scores had the most therapeutic potential for neurological disorders, followed by immunological disorders, cancer, pain, viral, and bacterial diseases. Several studies have also reported drugs with antiviral, anticancer, and anti-inflammatory properties for targeting SARS-CoV2 proteins [77]. This was then followed by scaffold analysis of the 162 drugs present in the top 10 drugs across the 24 SARS-CoV2 targets. Four common scaffolds (6,7,8,9-Tetrahydro-5H-cyclohepta[c]pyridine, 1-Benzazepine, decalin and leucoline) in level 2 and two common scaffolds (4,5,6,7,8,8a,9,10-Octahydro-2(3H)-phenanthrenone, Gona-1,3,5(10)-trien-3-ol) in level 3 of the scaffold tree were identified. The identified scaffolds along with the interacting residues of proteins that are involved in the interaction can be used to develop novel inhibitors with higher binding affinity. Since it is of paramount importance to find drugs that can bind to multiple targets in order to counteract viral mutational modifications, a polypharmacological property analysis was carried out where the top ten FDA approved drugs were filtered based on their interactions with more than three SARS-CoV2 target proteins. A total of 15 drugs were found to interact with multiple SARS-CoV2 targets, with dihydroergotamine and ergotamine leading the list by binding with eight targets. Then five number SAR-CoV2 targets were shown to interact with bisdequalinium chloride, midostaurin, temoporfin, tirilazad, and venetoclax. The original therapeutic indications of the identified drugs are quite varied, ranging from migraine, bacterial infection, neurological disorder and cancer. These multi-target drugs and their bound complex structure with the corresponding proteins resulted in identifying the key interacting residues involved in drug-target binding. ILE, LEU, ALA, ARG, and VAL are some of the most frequently occurring amino acid residues in the overall interactions, which are mostly hydrophobic in nature and contribute to the complexes' binding affinity. Among non-covalent bonds, hydrogen bonds play an important role in maintaining the stability of drug-target interactions, and according to the study, polar amino acids such as SER, ASN, GLN, and THR are majorly found to be involved in hydrogen formation. Targeting these residues will not only aid in comprehending the therapeutic effect of the drug on the target but will also contribute to developing high binding and target specific inhibitors. In the context of drug repurposing, pathway analysis can offer valuable insights into the mechanism of action of a drug. Therefore, in order to identify the functional category of key biological pathways implicated among the targets of the 15 drugs, a gene enrichment analysis of targets related to the 15 drugs that interact with multiple targets was conducted. Majority of the top 10 drugs mostly influence pathways which are related to basic biological functions such as cell cycle regulation, cell functionality, cell signalling, transmission, ion channelling, DNA repair, apoptosis and immune response. Furthermore, studies indicating that the identified pathways are linked to various biological process during SARS-CoV2 infections supports that these pathways and involved genes are important

drug repurposing targets and could be used for the development of drugs for COVID-19. In addition, the KEGG pathway analysis have led to the identification of important genes involved in a specific pathway and how the expression of gene is influenced under different condition particularly in relation to the SARS-CoV2 infection. The gene enrichment analysis and the KEGG pathways of genes connected to multi-target drugs reveals details about several pathways that could help in understanding the mechanisms of action underlying these drugs and how they could be repurposed against SARS-CoV2 targets. Drugs obtained through virtual screening with long term efficacy can be challenging in drug discovery process as virus undergo rapid mutation effecting the binding affinity and efficacy of drug. However, several studies have reported using a structure-based virtual screening strategy to find suitable alternative drugs for those that have developed resistance to SARS-CoV2 variants [78, 79]. One such strategy is the multi-target approach which is employed in the current study to identify approved drugs with polypharmacological properties. In the current study, FDA approved drugs were screened through a unique combination of docking study, therapeutic indication analysis along with polypharmacology and network pharmacology approach. This approach has led to identification of drugs targeting multiple proteins and pathways which is an essential component to enhance the efficacy of drug against viral variants. The traditional approaches of one target for one drug may become inefficient when the targets undergo frequent mutations. Besides considering the large number of pathways in viral entry, progression, replication, translocation, etc., a drug which can simultaneously inhibit multiple targets appear to be an optimal choice. Thus, the current study adopts polypharmacology assisted by network pharmacology.

## Conclusions

In the current study, virtual screening of FDA approved drugs was carried out against 24 SARS-CoV2 targets along with categorization of top docking scored drugs based on their therapeutic indications through a combination of polypharmacological and network pharmacology approaches. The common scaffolds generated by molecules with high docking scores are helpful in designing drugs for disrupting the virulence of SARS-CoV2, due to their simultaneous involvement in inhibiting multiple pathways. Further, analysing the gene enrichment and KEGG pathway analysis of genes related to multi target drugs, has led to the identification of various genes and biological pathways that can be targetable for drug repurposing. The repurposed drugs with original indications against neurological disorders (tirilazad), pain (dihydroergotamine, ergotamine), cancer (midostaurin, venetoclax, temoporfin) and bacterial infections (bisdequalinium chloride) may emerge as effective anti SARS-CoV2 drugs, owing to their multi-targeting nature.

## Supporting information

**S1 File.**
(DOCX)

## Acknowledgments

EJ thanks DST for Inspire Fellowship, KK thanks UGC for SRF Fellowship and HS thanks DBT Centre of Excellence for the fellowship.

## Author Contributions

**Conceptualization:** G. Narahari Sastry.

**Formal analysis:** Esther Jamir, Himakshi Sarma, Lipsa Priyadarsinee, Kikrusenuo Kiewhuo, Selvaraman Nagamani.

**Funding acquisition:** G. Narahari Sastry.

**Investigation:** G. Narahari Sastry.

**Methodology:** Esther Jamir, Himakshi Sarma, Lipsa Priyadarsinee, Kikrusenuo Kiewhuo, Selvaraman Nagamani.

**Supervision:** G. Narahari Sastry.

**Validation:** Selvaraman Nagamani, G. Narahari Sastry.

**Writing – original draft:** Esther Jamir.

**Writing – review & editing:** G. Narahari Sastry.

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
