## [Decision Letter · Decision Letter 0]

16 May 2023

PONE-D-23-10719Polypharmacology Guided Drug Repositioning Approach for SARS-CoV2PLOS ONE

Dear Dr. Jamir,

Thank you for submitting your manuscript to PLOS ONE. After careful consideration, we feel that it has merit but does not fully meet PLOS ONE’s publication criteria as it currently stands. Therefore, we invite you to submit a revised version of the manuscript that addresses the points raised during the review process.

We look forward to receiving your revised manuscript.

Kind regards,

Chandrabose Selvaraj, Ph.D.

Academic Editor

PLOS ONE

Journal Requirements:

3. Please note that PLOS ONE has specific guidelines on code sharing for submissions in which author-generated code underpins the findings in the manuscript. In these cases, all author-generated code must be made available without restrictions upon publication of the work. Please review our guidelines at https://journals.plos.org/plosone/s/materials-and-software-sharing#loc-sharing-code and ensure that your code is shared in a way that follows best practice and facilitates reproducibility and reuse

"GNS thanks DST New Delhi for the J.C Bose fellowship. EJ thanks DST for Inspire Fellowship, KK thanks UGC for SRF Fellowship and HS thanks DBT Centre of Excellence for the fellowship. The DBT is thanked for the funding in the form of Centre of Excellence in Advanced Computation and Data Sciences (No. BT/PR40188/BTIS/137/27/2021)."

"Funding to carry out the research work was obtained by DBT Centre of Excellence in Advanced Computation and Data Sciences (No. BT/PR40188/BTIS/137/27/2021)."

Reviewers' comments:

Reviewer's Responses to Questions

**Comments to the Author**

1. Is the manuscript technically sound, and do the data support the conclusions?

Reviewer #1: No

Reviewer #2: Yes

Reviewer #3: Yes

2. Has the statistical analysis been performed appropriately and rigorously? 

Reviewer #1: I Don't Know

Reviewer #2: Yes

Reviewer #3: No

3. Have the authors made all data underlying the findings in their manuscript fully available?

Reviewer #1: Yes

Reviewer #2: Yes

Reviewer #3: Yes

4. Is the manuscript presented in an intelligible fashion and written in standard English?

Reviewer #1: Yes

Reviewer #2: Yes

Reviewer #3: Yes

5. Review Comments to the Author

Reviewer #1: This manuscript is not sufficient enough to consider for publication; although it is close. Given the sheer amount of published work in the area, we are expecting not only simple docking studies but also docking results using at least three different programs and also multiple, replicated in at least triplicate 100ns-scale MD simulations with MM-GBSA/PBSA as well as a detailed and rigorous analysis including detailed structural analysis. A detailed discussion in the manuscript as to how the work is unique given the amount of research already done in this area is also necessary.

Reviewer #2: Drug repurposing is becoming the most attractive approach for identifying drug molecules/lead compounds against various viral and bacterial infections. Given that the multi-drug resistance is often seen in many microorganisms and the emergence of different new variants of viruses, this can be a pragmatic approach. In general, drug discovery involves a period of 13-15 years, and such an approach may not be effective in handling fastly emerging variants of novel coronaviruses. Another advantage of such drug repurposing is that these compounds already have favorable ADMET properties. Esther et al contributed with a drug repurposing study for 24 different proteins of SARS-CoV-2 which are selected based on the function, active site residues and binding pocket size. Among these, the structures were obtained from PDB for the 20 targets while for the remaining I-TASSER server has been used. The authors have considered 4193 FDA approved drugs for this study which were obtained from DrugCentral and DrugBank databases. The virtual screening against all these targets was carried out using AutodockVina software. In addition, for the top compounds Molecular Transfer Drug Target Interaction (MT-DTI) prediction which is a deep learning-based approach, has been used. In particular, this model uses the protein sequence information for the target and SMILES for the ligands for predicting the binding affinities. Interestingly, the binding scores predicted from MT-DTI were comparable to the docking energies from vina which validates the reliability of the latter approach. The sequential scoring using VINA and MT-DTI as employed in this study has also identified many previously reported COVID-19 drugs such as remdesivir, Baricitinib, ritonavir, hydroxychloroquine, Lopinavir, favipiravir and dexamethasone which are under clinical trial. This clearly indicates that the compounds obtained from such a sequential scoring can be lead compounds which further need to be validated through clinical trials. The list of top compounds has been further used to identify the privileged scaffold. In addition, the authors list a set of compounds that bind to more than 3 three targets of SARS-CoV-2. The multi-targeting drugs were shown effective against multi-drug resistant variants, and so this list of compounds can be considered for further validation in the clinical environment. Following are the remarks/comments about this manuscript.

(i) In my opinion, the study is well organized, and appropriate tools were used to identify the multi-targeting compounds from DrugBank/DrugCentra for treating Covid-19.

(ii) Through sequential scoring using VINA and MT-DTI. The study is also able to identify many lead compounds which are under clinical trial for treating the disease.

(iii) As far as I know, this is a unique study where about 24 different targets of SARS-CoV-2 were used to develop multi-targeting drug compounds.

The work is very timely and I strongly recommend the publication of this article in its present form. The quality of the selected figures (such as Figure 6 and 7) needs to be improved before publication

Reviewer #3: Review Comments

Abstract

• The abstract needs to be rewritten highlighting the background, methods employed, significant findings and conclusion.

• In line 28, ‘NSP1-NSP16’ can be changed into ‘NSP1-10 and NSP12- 16, envelope, membrane….ORF96’

Materials & Methods

• In line 108, ‘In the study’ can be changed into ‘In this study….(41)’

• In line 135, ‘The virtual …24SARS-COV2 target was performed using Autodock 1.12.’ shall be changed into ‘The virtual… 24SARS-COV2 targets was performed using Autodock Vina 1.1.2.’

• Why authors have not performed Molecular simulation studies?

Results

• In line 198, ‘The docking score and binding scores calculated … indications’ can be changed into ‘The docking and binding scores calculated….indications’.

• In line 234, ‘And drugs such as desloratadine …….. agents have obtained the lowest docking score of around -5.5 kcal/mol with NSP7 protein’ can be changed into ‘And drugs such as desloratadine ……agents have the lowest docking score of around -5.5 kcal/mol with NSP7 protein’.

• In line 279, ‘….. envelope and ORF7a protein’ can be changed into ‘….envelope and ORF7a proteins’.

• In the 280, ‘Several …….. remdesivir an antiviral drug …….’ can be changed into ‘Several ……. as an antiviral drug’.

• In line 305, ‘level 3 of the scaffold tree were observed and is shown in Fig 4’ can be changed into ‘level 3 of the scaffold tree were observed and are shown in Fig 4’.

• In line 333, ‘Among these drugs, dihydroergotaine and ergotamine drug showed…… targets’ can be changed into ‘Among these drugs, dihydroergotamine and ergotamine showed a maximum ……..targets.’

• In line 362, ‘In regard to this, it …….observed that amino acids which are polar in nature (65) such as SER, ASN, GLN, THR is ……..’ can be changed into ‘In regard to this, it can be observed that polar-amino acids (63) such as SER, ASN, GLN, THR are involved majorly in hydrogen formation’.

• In line 368, ‘the hydrophobic and hydrogen bond forming rsidues……..’ can be changed into ‘The hydrophobic and hydrogen bond forming residues are …….. in Fig S1.’

• In line 374, ‘Similarly, the drugs interacting with 5, 4 and 3 targets it was observed that few …… formation’ can be changed into ‘Similarly, among the drugs interacting with 5, 4 and 3 targets ….. formation.’

• In line 439, ‘While the pathway with ……. Pathway (Fig 7a)’ can be changed into ‘The pathway with the ………. signaling pathway (Fig 7a).’

• In line 449, ‘The plasma membrane ……… signal transmission (69)’ can be changed into ‘The plasma membrane is a highly ……. signal transmission (69).’

• In line 454, ‘so, it’s probable that ……. transducer activity (Fig 7c)’ can be changed into ‘so, it is probable that ……. transducer activity (Fig 7c).’

• In line 457, ‘This includes genes that encodes receptors…… signalling (69)’ can be changed into ‘This includes genes that encode receptors……. signaling (69).’

• In line 464, ‘This indicate the significance ……(67, 68)’ can be changed into ‘This indicates the significance………. (67, 68).’

• In line 497, ‘Each clusters are connected by edges …….’ can be changed into ‘Each cluster is connected by edges…….pathways.’

• In line 516, ‘In addition, the enrichment ……. genes mostly in disease related……’ can be changed into ‘In addition, the enrichment ….. mostly in diseases related to neurological ………. cancer.’

• A table for lead drugs with amino acid residues involved in the hydrogen bond formation and van der Walls interactions should be provided.

• HIV infections in Table 3(a) should be corrected as HIV infection.

• What does “CNS” mean in Tables? If CNS is “Central Nervous System”, it is not a therapeutic indication.

• Where from the therapeutic indications of drugs were captured? Cite the reference.

Discussion

• Since the SARS-CoV-2 functional genes continuously acquire mutations, how will the docked drugs be effective in future? This aspect should be discussed in the discussion section.

• In line 523, ‘The docking score of the top 10 FDA ……. Autodock Vina was validated ……’ can be changed into ‘The docking scores of the top 10 FDA …….Autodock Vina were validated using ……. both methods.’

• In line 528,’ It was found that the ………and infectious diseases such as viral, and bacterial diseases’ can be changed into ‘It was found that the ……. and infectious viral and bacterial diseases.’

Conclusion

• The conclusion is to be written better focusing only on the outcome of the study findings.

6. PLOS authors have the option to publish the peer review history of their article (what does this mean?). If published, this will include your full peer review and any attached files.

Reviewer #1: No

Reviewer #2: **Yes: **N. Arul Murugan

Reviewer #3: No

---

## [Author Response · Author response to Decision Letter 0]

9 Jun 2023

04 June 2023

To 

Dr. Chandrabose Selvaraj

Academic Editor, PLOS One 

Sub: Submission of the revised manuscript entitled “Polypharmacology Guided Drug Repositioning Approach for SARS-CoV2” (PONE-D-23-10719) by Jamir et. al.,

Dear Dr. Selvaraj, 

Thank you for your communication regarding the reviewer’s comment and considering our manuscript in the Plos One journal. We thank the reviewers for their constructive suggestion, which helped to improve the quality of the manuscript. Point-wise response follows:

Reviewer #1: 

Query 1. This manuscript is not sufficient enough to consider for publication; although it is close. Given the sheer amount of published work in the area, we are expecting not only simple docking studies but also docking results using at least three different programs and also multiple, replicated in at least triplicate 100ns-scale MD simulations with MM-GBSA/PBSA as well as a detailed and rigorous analysis including detailed structural analysis. A detailed discussion in the manuscript as to how the work is unique given the amount of research already done in this area is also necessary.

Author’s response:

a) We understand the reviewer’s concern and are thankful for the constructive comments. Following the suggestions of the reviewer, we tried to use two more docking programs and thus four different docking protocols are used in the revised version. The docking scores obtained from all the four programs were comparable which validates the docking scores of top 10 FDA approved drugs across the 24 SARS-CoV2 targets. Thus, we feel that the current study provides more strength to our study, as the results are essentially independent of the docking protocol employed. Regarding the molecular dynamics simulations, we would like to humbly submit that performing 100 ns MD simulations, for the 240 systems is prohibitive for our available computational resources. Besides, we would like to bring to your notice proteins and ligands considered in this study are of various size. While exhaustive MD simulations are quite insightful, the time step required for each system is different and that makes it not very practical to employ a uniform time step. The focus of the manuscript is on polypharmacology and drug repurposing, which will require some qualitative trends, as getting the accurate binding affinities appear to be difficult even with the MD simulations. Encouraged by the agreement among the various docking procedures and also considering the exhaustive nature of the complexes that need to be studied, we feel that employing MD simulations may not add much more deeper insights. 

The study in our opinion undertakes the most exhaustive approach for drug repurposing, by employing polypharmocology principles aided by therapeutic indication analysis and docking studies. This strategy has resulted in the identification of drugs that target several proteins and pathways, which are essential to increase a potency of drugs against the variants of the virus. The revised version presents these changes marked in yellow color.

(b) The uniqueness of the current study is that we employed a combination of docking study, therapeutic indication analysis, polypharmacological approach, and network pharmacology to screen FDA-approved drugs for repurposing against SARS-CoV2 targets. As suggested by the reviewer a detailed discussion on the uniqueness of the work carried out have been incorporated in the discussion section of the manuscript and highlighted in yellow colour.

Reviewer #2: 

Drug repurposing is becoming the most attractive approach for identifying drug molecules/lead compounds against various viral and bacterial infections. Given that the multi-drug resistance is often seen in many microorganisms and the emergence of different new variants of viruses, this can be a pragmatic approach. In general, drug discovery involves a period of 13-15 years, and such an approach may not be effective in handling fastly emerging variants of novel coronaviruses. Another advantage of such drug repurposing is that these compounds already have favorable ADMET properties. Esther et al contributed with a drug repurposing study for 24 different proteins of SARS-CoV-2 which are selected based on the function, active site residues and binding pocket size. Among these, the structures were obtained from PDB for the 20 targets while for the remaining I-TASSER server has been used. The authors have considered 4193 FDA approved drugs for this study which were obtained from DrugCentral and DrugBank databases. The virtual screening against all these targets was carried out using AutodockVina software. In addition, for the top compounds Molecular Transfer Drug Target Interaction (MT-DTI) prediction which is a deep learning-based approach, has been used. In particular, this model uses the protein sequence information for the target and SMILES for the ligands for predicting the binding affinities. Interestingly, the binding scores predicted from MT-DTI were comparable to the docking energies from vina which validates the reliability of the latter approach. The sequential scoring using VINA and MT-DTI as employed in this study has also identified many previously reported COVID-19 drugs such as remdesivir, Baricitinib, ritonavir, hydroxychloroquine, Lopinavir, favipiravir and dexamethasone which are under clinical trial. This clearly indicates that the compounds obtained from such a sequential scoring can be lead compounds which further need to be validated through clinical trials. The list of top compounds has been further used to identify the privileged scaffold. In addition, the authors list a set of compounds that bind to more than 3 three targets of SARS-CoV-2. The multi-targeting drugs were shown effective against multi-drug resistant variants, and so this list of compounds can be considered for further validation in the clinical environment. Following are the remarks/comments about this manuscript.

(i) In my opinion, the study is well organized, and appropriate tools were used to identify the multi-targeting compounds from DrugBank/DrugCentral for treating Covid-19.

(ii) Through sequential scoring using VINA and MT-DTI. The study is also able to identify many lead compounds which are under clinical trial for treating the disease.

(iii) As far as I know, this is a unique study where about 24 different targets of SARS-CoV-2 were used to develop multi-targeting drug compounds.

The work is very timely and I strongly recommend the publication of this article in its present form. 

Query 1. The quality of the selected figures (such as Figure 6 and 7) needs to be improved before publication

Author’s response: As suggested by the reviewer, we have enhanced the quality of figures 6 and 7. We have made a new Figure 6 and the previous Fig 6a-6d has been changed to Figure S4a-S4d and incorporated in supplementary the file. We hope that the improved Figures 6 and 7 are in acceptable form for publication.

 

Reviewer#3: 

Query 1. Abstract

• The abstract needs to be rewritten highlighting the background, methods employed, significant findings and conclusion.

• In line 28, ‘NSP1-NSP16’ can be changed into ‘NSP1-10 and NSP12- 16, envelope, membrane…ORF96’

Author’s response: As per the suggestions, the abstract has been rewritten highlighting the background, methods employed, significant findings and conclusion and is highlighted in yellow colour.

Query 2. Materials & Methods

• In line 108, ‘In the study’ can be changed into ‘In this study. (41)’

• In line 135, ‘The virtual …24SARS-COV2 target was performed using Autodock 1.12.’ shall be changed into ‘The virtual… 24SARS-COV2 targets was performed using Autodock Vina 1.1.2.’

Author’s response: We thank the reviewer for giving valuable suggestions which have been helpful in improving the manuscript. As suggested, all the modifications have been incorporated under Materials and Methods section of the manuscript and highlighted in yellow color.

Query 3. Why authors have not performed Molecular simulation studies?

Author’s response: We have addressed the complexity and prohibitively expensive nature of this task as a response to reviewer 1. As the focus of the manuscript on drug repurposing using polypharmacology approach has been exhaustively explored, the incremental benefit due to performing of the MD simulations was not attempted, especially when such a task is not possible to accomplish for us with our available computational resources. 

 

Query 4. Results

• In line 198, ‘The docking score and binding scores calculated … indications’ can be changed into ‘The docking and binding scores calculated…indications’.

• In line 234, ‘And drugs such as desloratadine …. agents have obtained the lowest docking score of around -5.5 kcal/mol with NSP7 protein’ can be changed into ‘And drugs such as desloratadine ……agents have the lowest docking score of around -5.5 kcal/mol with NSP7 protein’.

• In line 279, ‘…. envelope and ORF7a protein’ can be changed into ‘. envelope and ORF7a proteins’.

• In the 280, ‘Several ……. remdesivir an antiviral drug …….’ can be changed into ‘Several ……. as an antiviral drug’.

• In line 305, ‘level 3 of the scaffold tree were observed and is shown in Fig 4’ can be changed into ‘level 3 of the scaffold tree were observed and are shown in Fig 4’.

• In line 333, ‘Among these drugs, dihydroergotaine and ergotamine drug showed…… targets’ can be changed into ‘Among these drugs, dihydroergotamine and ergotamine showed a maximum …….targets.’

• In line 362, ‘In regard to this, it ……. observed that amino acids which are polar in nature (65) such as SER, ASN, GLN, THR is …….’ can be changed into ‘In regard to this, it can be observed that polar-amino acids (63) such as SER, ASN, GLN, THR are involved majorly in hydrogen formation’.

• In line 368, ‘the hydrophobic and hydrogen bond forming rsidues…….’ can be changed into ‘The hydrophobic and hydrogen bond forming residues are ……. in Fig S1.’

• In line 374, ‘Similarly, the drugs interacting with 5, 4 and 3 targets it was observed that few …… formation’ can be changed into ‘Similarly, among the drugs interacting with 5, 4 and 3 targets …... formation.’

• In line 439, ‘While the pathway with ……. Pathway (Fig 7a)’ can be changed into ‘The pathway with the ………. signaling pathway (Fig 7a).’

• In line 449, ‘The plasma membrane ……… signal transmission (69)’ can be changed into ‘The plasma membrane is a highly ……. signal transmission (69).’

• In line 454, ‘so, it’s probable that ……. transducer activity (Fig 7c)’ can be changed into ‘so, it is probable that ……. transducer activity (Fig 7c).’

• In line 457, ‘This includes genes that encodes receptors…… signalling (69)’ can be changed into ‘This includes genes that encode receptors……. signaling (69).’

• In line 464, ‘This indicate the significance …… (67, 68)’ can be changed into ‘This indicates the significance………. (67, 68).’

• In line 497, ‘Each clusters are connected by edges …….’ can be changed into ‘Each cluster is connected by edges……. pathways.’

• In line 516, ‘In addition, the enrichment ……. genes mostly in disease related……’ can be changed into ‘In addition, the enrichment …. mostly in diseases related to neurological ………. cancer.’

Author’s response: Many thanks to the reviewer for his suggestion that have been helpful in enhancing the quality the manuscript. We value the time and effort spent reading the manuscript carefully and providing helpful criticism. All the suggestions given by the reviewer have been taken into consideration and incorporated them in the result section of the manuscript and is highlighted in yellow color. 

Query 5. A table for lead drugs with amino acid residues involved in the hydrogen bond formation and van der Walls interactions should be provided.

Author’s response: As suggested by the reviewer, a table (Table S4) have been prepared with the top drugs interacting with multiple SARS-CoV2 targets along with the interacting residues and amino acid residues involved in the hydrogen bond formation and van der Waals interactions. The Table S4 have been incorporated in the supplementary information of the manuscript.

Query 6. HIV infections in Table 3(a) should be corrected as HIV infection.

Author’s response: As suggested, the HIV infections in Table 3(a) have been corrected as HIV infection.

Query 7. What does “CNS” mean in Tables? If CNS is “Central Nervous System”, it is not a therapeutic indication.

Author’s response: We are thankful to the reviewer for identifying the mistake in naming the therapeutic indication in the table. The CNS in tables have been now changed to CNS disorders which stands for Central Nervous System Disorders.

 

Query 8. Where from the therapeutic indications of drugs were captured? Cite the reference.

Author’s response: The therapeutic indication of each compound was retrieved from the drug databases Drug Bank and Drug Central database and the references have been cited which are references number 44 and 45. The particular sentence in mentioned and cited in the Materials and Methods section under the heading “Distribution of therapeutic indications toward the SARS-CoV2 targets” and also in result section under the heading “Therapeutic indications of the top 10 and bottom 10 drugs” which is highlighted in yellow colour.

Query 9. Discussion

• In line 523, ‘The docking score of the top 10 FDA ……. Autodock Vina was validated ……’ can be changed into ‘The docking scores of the top 10 FDA …. Autodock Vina were validated using ……. both methods.’

• In line 528,’ It was found that the ………and infectious diseases such as viral, and bacterial diseases’ can be changed into ‘It was found that the ……. and infectious viral and bacterial diseases.’

Author’s response: All the suggestions given by the reviewer have been taken into consideration and incorporated them in the discussion section of the manuscript and highlighted in yellow colour.

Query 10. Since the SARS-CoV-2 functional genes continuously acquire mutations, how will the docked drugs be effective in future? This aspect should be discussed in the discussion section.

Author’s response: It is an interesting point and we admit the limitations of the docking due to the possible loss of efficacy of drugs on mutation. This limitation is highlighted in the revised manuscript (page 24, line 567-582). However, this point raised by the referee is the root cause for us to explore the polypharmacology principle, as the mutations in multiple sites are far less probable.

 

Query 11. Conclusion

• The conclusion is to be written better focusing only on the outcome of the study findings.

Author’s response: The conclusion section has been rewritten focusing on the outcome of the study.

We hope the revised manuscript satisfactorily addressed all the queries raised by the reviewers. We thank the reviewers for their constructive comments and you for the prudent handling of the manuscript. 

Best Regards

G Narahari Sastry

---

## [Decision Letter · Decision Letter 1]

30 Jun 2023

PONE-D-23-10719R1Polypharmacology Guided Drug Repositioning Approach for SARS-CoV2PLOS ONE

Dear Dr. Jamir,

Thank you for submitting your manuscript to PLOS ONE. After careful consideration, we feel that it has merit but does not fully meet PLOS ONE’s publication criteria as it currently stands. Therefore, we invite you to submit a revised version of the manuscript that addresses the points raised during the review process.

We look forward to receiving your revised manuscript.

Kind regards,

Chandrabose Selvaraj, Ph.D.

Academic Editor

PLOS ONE

Reviewers' comments:

Reviewer's Responses to Questions

**Comments to the Author**

1. If the authors have adequately addressed your comments raised in a previous round of review and you feel that this manuscript is now acceptable for publication, you may indicate that here to bypass the “Comments to the Author” section, enter your conflict of interest statement in the “Confidential to Editor” section, and submit your "Accept" recommendation.

Reviewer #1: All comments have been addressed

Reviewer #3: All comments have been addressed

2. Is the manuscript technically sound, and do the data support the conclusions?

Reviewer #1: No

Reviewer #3: Yes

3. Has the statistical analysis been performed appropriately and rigorously? 

Reviewer #1: I Don't Know

Reviewer #3: (No Response)

4. Have the authors made all data underlying the findings in their manuscript fully available?

Reviewer #1: No

Reviewer #3: Yes

5. Is the manuscript presented in an intelligible fashion and written in standard English?

Reviewer #1: No

Reviewer #3: Yes

6. Review Comments to the Author

Reviewer #1: Although authors have responded to my request, docking studies without the support of MD simulations combined with end-point or alchemical free energy calculations are not enough to provide profound and strong computational drug repurposing or drug design results.

Reviewer #3: REVIEW COMMENTS

MINOR REVISION

I would like to submit following comments in connection with the manuscript titled “Polypharmacology Guided Drug Repositioning Approach for SARS-CoV2”.

Abstract

1. In lines no 30-40, As a result, the top 10 drugs ……….. biological process. Can be changed into “As a result, the top 10 drugs were found to have therapeutic indications for cancer, pain, neurological disorders, viral and bacterial diseases.

2. As drug resistance is one of the major challenges in antiviral drug discovery, polypharmacology and network pharmacology approaches were employed in this study to identify drugs interacting with multiple targets and drugs such as dihydroergotamine, ergotamine, bisdequalinium chloride, midostaurin, temoporfin, tirilazad, and venetoclax were identified among the multi-targeting drugs.

3. Further, a pathway analysis of the genes related to the multi-targeting drugs was carried out which provides insights into the mechanism of drugs and identifying targetable genes and biological pathways involved in SARS-CoV2.

Table

4. Table 3a, under Spike protein heading, HIV-1 infections can be changed into HIV-1 infection.

Results

5. In line no 441, According to a study reported by Deng et al., 2021, ……….plasma membrane [74]. Can be changed into “According to a study reported by Deng et al. (2021),, ………. and directs fusing of the viral membrane with the host plasma membrane [74].”

6. In lines no 454-456, This indicates the significance………..SARS-CoV-2 infection [71, 72] can be changed into “This indicates the significance………..SARS-CoV-2 infection [71, 72].”

Discussion

7. In line no 513, The docking score of the top 10 FDA approved………. all the three methods. Can be changed into “The docking scores of the top 10 FDA approved………. all the three methods.”

This manuscript can be considered for publication after carrying out the suggestions pointed out.

7. PLOS authors have the option to publish the peer review history of their article (what does this mean?). If published, this will include your full peer review and any attached files.

Reviewer #1: No

Reviewer #3: No

---

## [Author Response · Author response to Decision Letter 1]

19 Jul 2023

Dr. Chandrabose Selvaraj

Academic Editor, PLOS One 

Sub: Submission of the revised manuscript entitled “Polypharmacology Guided Drug Repositioning Approach for SARS-CoV2” (PONE-D-23-10719) by Jamir et. al.,

Dear Professor Chandrabose, 

Thank you for your communication regarding the reviewer’s comment and considering our manuscript our revised manuscript. Current second revision contain makes an attempt to satisfactorily address the queries raised by the reviewers. The changes are incorporated in the manuscript and highlighted. You please find the point-wise response below, 

Reviewer #1

Query 1. Although authors have responded to my request, docking studies without the support of MD simulations combined with end-point or alchemical free energy calculations are not enough to provide profound and strong computational drug repurposing or drug design results.

Answer. While we appreciate the concern of the referee regarding the rigorous quantification of the binding energy some of our practical concerns, we would like to bring forth our apprehensions as given in the earlier response to the query in revision 1. 

While exhaustive MD simulations are quite insightful, the time step required for each system is different and that makes it not very practical to employ a uniform time step. Further, we would like to bring to the attention of the revered referee that proteins and ligands considered in this study are of various size and getting a consistent and comparable trajectory will be a challenge. In addition to the three docking procedures that we have adopted, we chose to search for a reliable alternative to the docking method. Thus, we employed MM-PB/GBSA analysis using the web server http://cadd.zju.edu.cn/fastdrh/submit [Briefings in Bioinformatics, 23(5), bbac201], while these approaches are far from traditional MD trajectory driven analogs, the MM-PB/GBSA obtained are in good agreement with the docking results.

Further, we also would like to mention that the focus of the current manuscript is on polypharmacology and drug repurposing, which are essentially dependent on qualitative trends. Given the consensus observed among different docking procedures and the comprehensive nature of the complexes under investigation, we believe that incorporating MD simulations may not provide significantly deeper insights. In our opinion, this study adopts the most exhaustive approach to drug repurposing by utilizing polypharmacology principles, aided by therapeutic indication analysis and docking studies. This strategy has successfully identified drugs that target multiple proteins and pathways critical for enhancing drug potency against virus variants. 

 

Reviewer #3: 

MINOR REVISION

I would like to submit following comments in connection with the manuscript titled “Polypharmacology Guided Drug Repositioning Approach for SARS-CoV2”.

Abstract

Query 1. In lines no 30-40, As a result, the top 10 drugs ……….. biological process. Can be changed into “As a result, the top 10 drugs were found to have therapeutic indications for cancer, pain, neurological disorders, viral and bacterial diseases.

Answer. We thank the reviewer for taking time to go through the manuscript and providing valuable suggestions. As suggested, the line “As a result, the top 10 drugs ……….. biological process” have been changed into “As a result, the top 10 drugs were found to have therapeutic indications for cancer, pain, neurological disorders, viral and bacterial diseases.

Query 2. As drug resistance is one of the major challenges in antiviral drug discovery, polypharmacology and network pharmacology approaches were employed in this study to identify drugs interacting with multiple targets and drugs such as dihydroergotamine, ergotamine, bisdequalinium chloride, midostaurin, temoporfin, tirilazad, and venetoclax were identified among the multi-targeting drugs.

Answer. The sentence has been modified as per the suggestion of the reviewer.

Query 3. Further, a pathway analysis of the genes related to the multi-targeting drugs was carried out which provides insights into the mechanism of drugs and identifying targetable genes and biological pathways involved in SARS-CoV2.

Answer. The sentence has been modified as per the suggestion of the reviewer.

Table

Query 4. Table 3a, under Spike protein heading, HIV-1 infections can be changed into HIV-1 infection.

Answer. Table 3a, under Spike protein heading, HIV-1 infections have be changed into HIV-1 infection.

Results

Query 5. In line no 441, According to a study reported by Deng et al., 2021, ……….plasma membrane [74]. Can be changed into “According to a study reported by Deng et al. (2021),, ………. and directs fusing of the viral membrane with the host plasma membrane [74].”

Answer. The line, “According to a study reported by Deng et al., 2021, ……….plasma membrane [74]. have been changed into “According to a study reported by Deng et al. (2021),, ………. and directs fusing of the viral membrane with the host plasma membrane [74].”

Query 6. In lines no 454-456, This indicates the significance………..SARS-CoV-2 infection [71, 72] can be changed into “This indicates the significance………..SARS-CoV-2 infection [71, 72].” 

Answer. The line “This indicates the significance………..SARS-CoV-2 infection [71, 72] have been changed into “This indicates the significance………..SARS-CoV-2 infection [71, 72].” 

Discussion

Query 7. In line no 513, The docking score of the top 10 FDA approved………. all the three methods. Can be changed into “The docking scores of the top 10 FDA approved………. all the three methods.”

Answer. The line “The docking score of the top 10 FDA approved………. all the three methods have been into “The docking scores of the top 10 FDA approved………. all the three methods.”

We hope that our revision is satisfactory for you. We remain with many thanks to the reviewers for their constructive suggestions and you for the prudent handling of the manuscript. 

With Regards,

G. Narahari Sastry

---

## [Decision Letter · Decision Letter 2]

28 Jul 2023

Polypharmacology Guided Drug Repositioning Approach for SARS-CoV2

PONE-D-23-10719R2

Dear Dr. Jamir,

We’re pleased to inform you that your manuscript has been judged scientifically suitable for publication and will be formally accepted for publication once it meets all outstanding technical requirements.

Kind regards,

Chandrabose Selvaraj, Ph.D.

Academic Editor

PLOS ONE

Additional Editor Comments (optional):

Reviewers' comments:

Reviewer's Responses to Questions

**Comments to the Author**

1. If the authors have adequately addressed your comments raised in a previous round of review and you feel that this manuscript is now acceptable for publication, you may indicate that here to bypass the “Comments to the Author” section, enter your conflict of interest statement in the “Confidential to Editor” section, and submit your "Accept" recommendation.

Reviewer #3: All comments have been addressed

2. Is the manuscript technically sound, and do the data support the conclusions?

Reviewer #3: Yes

3. Has the statistical analysis been performed appropriately and rigorously? 

Reviewer #3: N/A

4. Have the authors made all data underlying the findings in their manuscript fully available?

Reviewer #3: Yes

5. Is the manuscript presented in an intelligible fashion and written in standard English?

Reviewer #3: Yes

6. Review Comments to the Author

Reviewer #3: I would like to submit the following comments with the manuscript titled “Polypharmacology Guided Drug –Repositioning Approach for SARS-CoV2”.

1. The authors responded to all my queries and also the queries of reviewer 1.

2. The manuscript in the present form can be considered for publication.

7. PLOS authors have the option to publish the peer review history of their article (what does this mean?). If published, this will include your full peer review and any attached files.

Reviewer #3: No

---

## [Editor Report · Acceptance letter]

1 Aug 2023

PONE-D-23-10719R2 

Polypharmacology Guided Drug Repositioning Approach for SARS-CoV2 

Dear Dr. Jamir:

I'm pleased to inform you that your manuscript has been deemed suitable for publication in PLOS ONE. Congratulations! Your manuscript is now with our production department. 

Kind regards, 

on behalf of

Dr. Chandrabose Selvaraj 

Academic Editor

PLOS ONE